# Do Bacteria Provide an Alternative to Cancer Treatment and What Role Does Lactic Acid Bacteria Play?

**DOI:** 10.3390/microorganisms10091733

**Published:** 2022-08-27

**Authors:** Leon M. T. Dicks, Wian Vermeulen

**Affiliations:** Department of Microbiology, Stellenbosch University, Private Bab X1, Matieland, Stellenbosch 7602, South Africa

**Keywords:** bacteria, lactic acid bacteria, cancer treatment

## Abstract

Cancer is one of the leading causes of mortality and morbidity worldwide. According to 2022 statistics from the World Health Organization (WHO), close to 10 million deaths have been reported in 2020 and it is estimated that the number of cancer cases world-wide could increase to 21.6 million by 2030. Breast, lung, thyroid, pancreatic, liver, prostate, bladder, kidney, pelvis, colon, and rectum cancers are the most prevalent. Each year, approximately 400,000 children develop cancer. Treatment between countries vary, but usually includes either surgery, radiotherapy, or chemotherapy. Modern treatments such as hormone-, immuno- and antibody-based therapies are becoming increasingly popular. Several recent reports have been published on toxins, antibiotics, bacteriocins, non-ribosomal peptides, polyketides, phenylpropanoids, phenylflavonoids, purine nucleosides, short chain fatty acids (SCFAs) and enzymes with anticancer properties. Most of these molecules target cancer cells in a selective manner, either directly or indirectly through specific pathways. This review discusses the role of bacteria, including lactic acid bacteria, and their metabolites in the treatment of cancer.

## 1. Introduction

Cancer is one of the leading causes of death, according to 2022 statistics published by the American Cancer Society [1]. At the 17th World Health Assembly [2], the WHO passed the resolution “cancer prevention and control in the context of an integrated approach” and predicted that the number of cancer cases world-wide could increase to 21.6 million by 2030. The American Cancer Society predicted 1.9 million new cancer cases and 609 360 deaths in the United States of America in 2022, rating cancer as the second leading cause of death [1]. According to the latest statistics from the South African National Cancer Registry (NCR), published by the National Institute for Communicable Diseases (NICD), 85 302 new cancer cases have been reported in 2019 [3]. Of all cancers, breast, lung, prostate, colon, rectum, bladder, kidney, renal, pelvis, pancreatic, thyroid and liver cancer are most common [4].

Cancer is normally treated with radiotherapy or chemotherapy and tumours are surgically removed. Less severe cancers are treated with hormone-, immune- and antibody-based therapies [5]. Most of these treatments, however, do not discriminate between normal cells and cancer cells, and vary concerning the ability to infiltrate tumours. Patients that receive radiotherapy or chemotherapy complain of adverse side effects such as flu-like symptoms, heart problems, diarrhoea, nausea, lockjaw (trismus) and chronic bladder spams [6,7,8,9]. Radiotherapy of the neck and head may elicit difficulty in swallowing (dysphagia), dry mouth feel (xerostomia), necrosis, inflammation of the spinal cord, and even permanent trismus or neurological damage [6,10]. Approximately 90% of patients with advanced cancer experience severe pain after surgery [11] and need to take opioids. This may lead to drug abuse [12]. In some cases, cancer cells have developed resistance to conventional treatments [13], which emphasizes the need to search for alternative treatments and anticancer drugs. This led to the search for anticancer compounds produced by plants, marine organisms, fungi, algae, and bacteria [14]. Several toxins, antibiotics, bacteriocins, non-ribosomal peptides, polyketides, phenylpropanoids, phenylflavonoids, purine nucleosides, short chain fatty acids (SCFAs) and enzymes with anticancer properties, mostly produced by bacteria, have been described [15]. The challenge is to identify compounds that only target cancer cells and not normal cells.

Cancer cells differ from normal cells by having more fluidic cell membranes [16], a net negative charge (due to elevated levels of phosphatidylserine, O-glycosylated mucins, sialylated gangliosides and heparin sulfates), and more microvilli, thus a larger surface area [17,18,19,20]. Because of these characteristics, cationic peptides such as bacteriocins may find it easier to adhere to cancer cells than normal cells [21,22].

The aim of this review is not to discuss various cancers and symptoms, nor the advantages and disadvantages of conventional therapies, but to provide the reader with the latest developments in the use of bacterial toxins, antibiotics, bacteriocins, non-ribosomal peptides, polyketides, phenylpropanoids, phenylflavonoids, purine nucleosides, SCFAs and enzymes in cancer treatment. The possibility of using lactic acid bacteria or their bacteriocins as anticancer agents is investigated.

## 2. Bacterial-Mediated Cancer Therapy

Experiments with viable or attenuated microorganisms to treat cancer dates back more than a century [7,23,24]. The first cancer vaccine, composed of viable *Streptococcus pyogenes* cells, was developed by Dr Coley in 1891 [23,24,25]. The cells activated macrophages and lymphocytes, and stimulated production of tumour necrosis factor α (TNFα) that regulates inflammatory responses required to attack malignant neoplasm [26,27]. The “Coley-toxin”, that consisted of heat-treated culture supernatants of *S. pyogenes* and *Serratia marcescens*, was rejected by the USA Food and Drug Administration (FDA) in 1962 due to lack of unequivocal scientific support and reports of organ damage [23,28]. Subsequent reports of *Clostridium*, *Corynebacterium*, *Bacillus* Calmette-Guérin, *Salmonella*, *Escherichia coli*, *Bifidobacterium* and *Listeria* associated with cancer cells [29,30,31,32,33] led to renewed interest in the search for bacterial cells with anticancer properties, especially species autochthonous to the human gut. Several papers have been published advocating the use of *Salmonella* in cancer treatment.

*Salmonella typhimurium* VNP20009 was made less toxic by deleting *msbB* encoding lipopolysaccharide (LPS) production [34]. By deleting *purI* the strain became auxotrophic for adenine [35]. In a later study, King et al. [36] engineered strain VNP20009 to express cytosine deaminase (CD) that converts 5-fluorocytosine (5-FC) to 5-fluorouracil (5-FU). Fluorouracil, commercially known as Adrucil, is a cytotoxin used in treatment of colorectal, oesophageal, stomach, pancreatic, breast, and cervical cancers [37]. In murine models 5-FU formed at tumour sites, resulting in a dramatic repression of cell growth [36,38]. A CD-expressing strain of *Salmonella enterica* yielded similar results when tested in mice [39], raising the hope that engineered strains of *Salmonella* may be used to activate cytotoxic drugs within tumour cells. Another strain of *S. typhimurium* (TAPET-CD, also referred to as VNP20029) expressed genes encoding CD and colonized tumour cells for at least 15 days. The strain converted 5-FC to 5-FU in 2 of the 3 patients treated [40]. No reports of follow-up clinical trials with any of these strains have been reported.

Inducible promoters responding to specific conditions in tumour cells (e.g., hypoxia) were used to design delivery systems for anti-cancer treatment [41]. The hypoxia-inducible promoter (HIP) and a fluorescent marker were cloned into *S. typhimurium* VNP20009 [42]. A 15-fold increase in fluorescence was recorded in HCT116 human colorectal carcinoma cells. Non-cancerous cells did not fluoresce [42], which suggested the *Salmonalla* delivery vector is very specific. Other HIPs experimented with included those regulating the expression of *pflE* and *ansB* [43]. Swofford et al. [44] transformed the luxI/luxR quorum-sensing system of *Vibrio fischeri* into *S. typhimurium* VPN200010. These reporter genes were expressed in 4T1 mammary tumours, but only when cell densities were above a certain threshold.

Other strains of *S. typhimurium* tested for anti-cancer therapy included strain AR-1, an arginine and leucine auxotroph defective in ppGpp synthesis (strain ΔppGpp) [45], and strain SF200 with mutations in lipid A and flagella synthesis [46]. Clinical trials on metastatic cancer patients intravenously injected with maximum tolerable cell numbers of strain VNP20009 (3 × 10^8^ cfu/m^2^) proved ineffective [47]. Higher cell numbers led to anaemia, low platelet counts, bacteraemia, high blood bilirubin, nausea, vomiting, diarrhoea, hypophosphatemia, and an increase in alkaline phosphatase [47]. A genetically engineered strain of VNP20009 that produced TNF-related apoptosis-inducing ligand (TRAIL) under control of a γ-irradiation-inducible RecA promoter, stimulated caspase-3-mediated apoptosis in 4T1 mammary carcinoma cells [48]. *S. typhimurium* ΔppGpp that expressed tissue inhibitor of metalloproteinases 2 (TIMP-2) reduced the size of glioma brain tumours in BALB/c mice and increased survival by 60% [49]. Modification of *S. typhimurium* ΔppGpp to express the mitochondrial targeting domain of Noxa (MTD), fused with the cell-penetrating peptide DS4.3 (DS4.3-MTD), led to the complete destruction of colon carcinoma tumours [50]. An engineered strain of *Salmonella choleraesuis* that expressed the angiogenesis inhibitor thrombospondin-1 inhibited the growth of B16F10 melanoma cells in mice [51]. No clinical trial data have been published on any of these genetically engineered strains.

A strain of *S. typhimurium* genetically engineered to produce truncated human interleukin-2 (SalpIL2), reduced adenocarcinoma metastases of the liver [52,53,54,55]. Sorensen et al. [56] reported that metastatic osteosarcoma in a mouse model could be treated with a single oral dose of attenuated SalpIL2-producing *S. typhimurium*. Barnett et al. [57] have shown that SalpIL2 reduced the volume and mass of retroperitoneal neuroblastoma tumours in a murine model. In vitro experiments have shown that SalpIL2-producing strains invade and divide within K7M2 osteosarcoma cells [58], suggesting that SalpIL2 may persist for long periods in malignant tissue.

Significant lysis of tumor cells in mice were recorded with *Clostridium histolyticum* treatment [58]. Shrinking of tumor cells were observed when these cells were exposed to *Clostridium tetani* [59]. *Clostridium novyi*, made non-pathogenic by deleting the gene encoding α-toxin NT, destroyed tumours and secreted liposomase [60]. The latter has been experimented with in enhancing the release of liposome-encapsulated drugs within tumours [61]. Endospores of *C. novyi*-NT colonized the hypoxic regions of tumours, elicited cell lysis and an immune inflammatory response that resulted in immunogenic cell death [60,62]. In a phase I study, injection of *C. novyi*-NT into tumour cells led to a decrease in the size of the cells, resulted in a systemic cytokine response and enhanced systemic tumor-specific T-cell responses [63]. Bettegowda et al. [64] reported a long-term remediation effect when tumours in a mice model were treated with a combination of *C novyi*-NT endospores and radiation therapy. The authors suggested that a combination of radioactive iodine with *C novyi*-NT might enable patients to be treated with lower doses of radiolabeled antibodies, which limits injury to normal tissue such as bone marrow.

Nuyts et al. [65] has shown that the *recA* and *recN* genes in *Clostridium acetobutylicum* DSM792 could be activated with a radiation dose of 2 Gy. The authors argued that the activation of the recA promoter could increase TNFα production in recombinant clostridia. In a later study, Jiang et al. [66] showed that *E coli* K12, harbouring plasmid pAClyA, produced higher levels of cytolysin A, which enhanced the therapeutic effects of radiation.

Cloning the genes encoding nitroreductase from *E. coli* to *Clostridium beijerinckii* resulted in the activation of CB1954 (nontoxic) into a drug with anticancer properties. Intravenous injection of activated CB1954 into mice destroyed tumours [67]. *Clostridium* spp. engineered to deliver CD to tumours [68] have also been modified to secrete TNFα [69] and may, in future, be used in cancer therapy. Li et al. [70] suppressed the growth of Heps mouse liver cancer cells in vivo by using a genetically engineered strain of *Bifidobacterium adolescentis*. *Bifodobacterium* spp. have also been used to deliver active CD enzymes to hypoxic regions of solid tumours in mice [71,72,73,74].

The first report using *Listeria monocytogenes* to direct an immune response to tumours was published by Pan et al. [75]. The authors have genetically engineered *L. monocytogenes* to secrete influenza virus nucleoprotein and have shown that the protein could repress tumours in colon and renal cancer models. Several other model studies were published using recombinant strains of *Listeria* to repress cervical, head and neck, breast, skin, and renal cancers; reviewed by Guirnalda et al. [76] and Cory and Chu [77]. An immunotherapy-based treatment for cervical cancer was developed based on live attenuated *L. monocytogenes* that secretes the fusion protein *Lm*-LLO-E7 [78]. The protein, referred to as ADXS11-001 (ADXS-HPV), targets human papillomavirus (HPV)-associated tumours. The immune response elicited by the ADXS11-001 vaccine against HPV oncoprotein E7 led to the reduction in tumour cells in animal models. Phase I and II clinical studies were later conducted [79,80,81]. Clinical trials are being conducted to evaluate another genetically engineered vaccine, protein ADXS-504, for treatment of biochemically recurrent (early) prostate cancer (https://www.globenewswire.com/en/news-release/2020/09/24/2098634/0/en/Advaxis-Announces-FDA-Clearance-of-New-IND-for-ADXS-504-for-Treatment-of-Prostate-Cancer.html#:~:text=filingsmedia%20partners-,Advaxis%20Announces%20FDA%20Clearance%20of%20New%20IND%20for%20ADXS%2D504,%7C%20Source%3A%20Advaxis%2C%20Inc; https://www.globenewswire.com/news-release/2022/06/29/2471231/0/en/Advaxis-Updates-on-the-Phase-1-Clinical-Trial-of-ADXS-504-for-the-Treatment-of-Early-Prostate-Cancer.html, assessed on 15 August 2022).

*E. coli* Nissle 1917 was genetically engineered to convert NH3 to L-arginine (L-arg). The recombinant strain, referred to as SYNB1020, reduced systemic hyperammonemia in mouse models [82]. A phase 1 clinical study showed that SYNB1020 was well tolerated at daily doses of up to 1.5 × 10^12^ cfu (colony-forming units) administered for up to 14 days. An increase in urinary nitrate, plasma ^15^N-nitrate and urinary ^15^N-nitrate was reported, suggesting that SYNB1020 could be used to treat hyperammonemia, including urea cycle disorders and hepatic encephalopathy [82]. Another genetically engineered strain of *E. coli* Nissle 1917, strain SYNB1618, yielded promising results when tested for the ability to alleviate phenylketonuria (PKU), a disorder caused by defective phenylalanine hydrolase, thus the inability to convert phenylalanine (Phe) to tyrosine [83]. Dose-responsive increases were observed in plasma (trans-cinnamic acid) and urine (hippuric acid) levels of Phe metabolites, suggesting that genetically engineered *E. coli* may be used in the treatment of rare metabolic disorders [83].

Intestinal bacteria influence various inflammatory and immune processes, many of which are implicated in tumour etiology, such as in colorectal cancer (CRC) [84]. *Bacteroides fragilis* and *Fusobacterium* spp. are directly associated with tumours, including CRC [85,86]. *Fusobacterium nucleatum* suppresses the immune response that leads to the induction of chronic inflammation [85]. *Bacteroides fragilis* alters (damages) the DNA of host cells, increases cell proliferation, and induces pro-inflammatory processes through the production of toxins [86]. A gene encoding *B. fragilis* toxin detected in colonic mucosa is associated with late-stage CRC [86]. Further research is required to determine if these species could be genetically modified to prevent the proliferation of cancer cells.

Although gut microbiota may prevent CRC, they also pose a risk of inducing CRC. This is mostly diet related. High levels of secondary bile acids (BAs) are produced from a high fat content diet [87,88]. Abnormal high levels of BAs in the colon induces inflammation [89,90] and forms reactive oxygen species that disrupts cell membranes and mitochondria [88]. Species primarily responsible for production of BAs are *Clostridium scindens, Clostridium hiranonis, Clostridium hylemonae* and *Clostridium sordellii* [91]. A diet rich in proteins and low in carbohydrates my also cause CRC, as reported by Russel et al. [92]. Fermentation of proteins in the distal colon leads to the production of toxic ammonia, amines, phenols and sulfides [93]. Lithocholic acid (LCA), a derivative of cholic acid, is an exception to the rule, as it inhibits the growth of human prostate cancer cells LNCaP and PC-3 by induction of caspase-3, 8 and 9 mediated apoptosis [94]. LCA not only induces endoplasmic reticulum (ER) stress that triggers the unfolded protein response (UPR) activating cell death [95], but transforms growth factor-β in HepG2 liver cancer cells, and suppresses the growth of breast cancer cells [96]. In addition, LCA induces oxidative phosphorylation, inhibits epithelial-mesenchymal transition and expression of vascular endothelial growth factor A, and stimulates antitumor immunity [96].

The anticarcinogenic properties of lactic acid bacteria (LAB) is addressed in far fewer publications and focuses mainly on exopolysaccharides (EPS), peptidoglycan, nucleic acid, bacteriocins, and S-layer proteins [97]. Viable cells of *Lactobacillus casei* BL23, intranasally administered using the human pappilomavirus (HPV)-induced model, reduced tumour growth [98]. *Lactobacillus reuteri* BCRC14652, tested in vitro, damaged the cell membranes of colon carcinoma HT29 cells [99], suppressed tumor necrosis factor (TNF)-induced NF-κB activation, and repressed the growth of cancer cells by apoptosis [100]. EPS produced by *Lactobacillus acidophilus* and *Lactobacillus rhamnosus* repressed the growth of HT-29 cells by inducing the activity of Beclin-1 (an autophagy protein) and GRP78 (an endoplasmic reticulum chaperone) directly, and indirectly by regulating apoptosis through stimulation of *Bcl-2* (B-cell lymphoma 2) and *Bak*, a pro-apoptotic gene of the Bcl-2 family [101]. A combination of *L. acidophilus* and *L. casei*, used with 5-FU, induced apoptosis of LS513 cancer cells [102], suggesting that these species may be used as adjuvants in anticancer chemotherapy.

Anti-tumor activities were also reported for cell-free supernatants of LAB, and irradiation-inactivated and heat-killed cells of LAB [99,102,103,104,105]. Exopolysaccharides (EPS) produced by *L. casei* 01 reduced the cytotoxicity of 4-nitroquinoline N-oxide (4-NQO), a pro-mutagen [106]. EPS isolated from *L. acidophilus* 606 repressed the growth of cancerous cells [103] and EPS from *Lactobacillus plantarum* and *L. acidophilus* significantly reduced tumour growth [107,108,109]. An interesting observation is that the repression of tumour growth exerted by an EPS-producing strain of *L. acidophilus* (strain LA1) may be associated with the suppression of lactate dehydrogenase (LDH) and alkaline phosphatase (ALP). Inhibition of LDH in the glycolytic pathway of cancer cells results in lower ATP production, hence slower growth [110,111]. EPS116, produced by *L. plantarum* NCU116, binds to TLR2 and activates the TLR2/MyD88/TRAF6/MKK7 pathway, which, in turn, activates JNK/c-Jun that upregulates the transcription and translation of *Fas* and *Fasl*. The Fas/Fasl signaling pathway activates FADD of caspase-8 and caspase-3. Activated Caspase-3 facilitates apoptosis by upregulating the expression of cellular target proteins PARPs and Rock1, followed by cleavage of PARP1 and inhibition of CT26 growth [112]. Many probiotic LAB produce EPS and may, in future, be used as alternative or complementary treatment of cancer. The anticancer properties of EPS are reviewed by Wu et al. [113].

Several reports highlighted the importance of LAB in the prevention of CRC, reviewed by Zhong [114]. The health and quality of life of patients that underwent surgical resection of CRC were significantly improved when administered *L. acidophilus* and *Bacillus natto* [115]. Despite this, there is no consensus on the role LAB play in CRC treatment. It is, however, certain that a select few LAB activate mechanisms involved in the killing or repression of cancer cells, and that they regulate immune response [116]. This includes neutralizing free radicals [117] and inactivation of reactive oxygen species (ROS) by NADH oxidase/peroxidase and catalase [118,119]. Strains of *Bifidobacterium longum* and *L. acidophilus* displayed antioxidative activity by inhibiting linoleic acid peroxidation [120]. Heat-killed cells of *L. acidophilus* 606 and EPS produced by the strain has potent antioxidative activity [103]. According to Kumar et al. [121] and Annuk et al. [122], obligate homofermentative lactobacilli display high antioxidant activity, but is highly strain-dependent among facultative and obligate heterofermentative lactobacilli.

LAB play an important role in stimulating the immune system, especially anti-inflammatory cytokine IL-10 [123]. Lipoteichoic acid, present in the cell walls of all LAB, stimulate DCs through Toll-like receptor 2, resulting in the release of cytokines [124,125]. Some lactobacilli stimulate DCs to produce IL-12 and IL-10 [126,127]. Disruption of LTA in *L. acidophilus* resulted in the production of IL-10 by DCs and the downregulation of IL-12, which led to T cell-mediated colitis in mice [128]. It thus seems possible to treat CRC by altering the cell surface components of *L. acidophilus*. A strain of *L. acidophilus* deficient in LTA (strain NCK2025) repressed the growth of colonic polyps by downregulating IL-12, TNF-α and IL-10. This activated CD4+ T-cells, as observed in a mice model [129]. In vivo studies have shown an increase in cytoplasmic levels of TNF-α, interferon-γ (IFN-γ) and IL-10 in animals administered *L. casei* and *B.*
*longum* [130]. This correlated with an increase in T cells, NK cells and MHC class II^+^ cells, and CD4-CD8^+^ T cells in a murine model [131]. *L. casei* Shirota (LcS) suppressed chemically induced carcinogenesis [131], supported by an increase in IFN-γ, interleukin-β (IL-1 β) and TNF-α levels [132]. A butanol extract prepared from the cell-free supernatant of *B.*
*adolescentis* significantly increased the production of TNF-α and NO, which regulated immune modulation and repressed tumor growth [133].

Messenger RNA (mRNA) and interferon gamma (IFN-γ) levels of leukemia KHYG-1 cells increased when treated with a combination of *Lactococcus lactis* subsp. *lactis*, *Lactococcus lactis* subsp. *cremoris*, *L. Lactococcus lactis* subsp. *lactis* biovar *diacetylactis*, *L. plantarum*, *Leuconostoc mesenteroides* subsp. *cremoris*, and *L. casei* [134]. The six LAB enhanced the cytotoxicity to human chronic myelogenous leukemia K562 cells and colorectal tumor HCT116 cells [134]. *L.*
*reuteri* ATCC-PTA-6475 reduced the growth of mammary tumors in Swiss mice by blocking NFκ-B-p65 nuclear translocation and the expression of c-jun, an oncogenic transcription factor [135]. Phenyllactic acid (PLA), hydroxyphenyllactic acid (OH-PLA), lactic acid, and indole lactic acid (ILA) produced by *L. plantarum* UM55 inhibited the growth of *Aspergillus flavus* and thus production of carcinogenic aflatoxins in food products [136].

Peptidoglycan isolated from *Bifidobacterium infantis* ATCC 15,697 repressed the growth of Meth A fibrosarcoma in BALB/c mice [137]. A cell wall-derived polysaccharide-peptidoglycan complex (PSPG) from *L. casei* Shirota prevented the activation of IL-6/STAT3 signalling and repressed ileal cancer [138]. Peptidoglycan from *Lactococcus* and *Bifidobacterium* inhibited the growth of bladder cancer HT-1376, colon cancer DLD-1 and SNUC2A cells, and kidney cancer A498 cells [139]. S-layer proteins from *L. acidophilus* CICC 6074 up-regulating the expression of p53, p21, and p16 and down-regulated the expression of CDK1 (cyclin-dependent kinase) and cyclin B in colon cancer HT-29 cells [140].

RNA extracted from the cell-free supernatant of *Lactobacillus* DM9811 inhibited the growth of colon cancer HT-29 cells and mouse ascites hepatoma cells [141]. The authors ascribed the anticarcinogenic activity to increased activity of NK and CD4^+^ T cells and the upregulation of cellular immunity. DNA fragments of *Lactobacillus bulgaricus* and *Streptococcus thermophilus* promoted mitosis in mouse spleen B and Pierre spot cells, resulting in enhanced immune functions [141]. These studies may be the first towards nucleic acid-based vaccines in cancer therapy.

Probiotic LAB and bifidobacteria inhibit signaling of epidermal growth factor receptor (EGFR) pathways [142], leading to an increase in phosphorylation of cytoplasmic tyrosine kinase domains. This causes the activation of cell proliferation, apoptosis, migration and differentiation [143].

Jackson Laboratory (JAX) mice, known to be easily colonized by commensal microbes, had higher cell numbers of *Bifidobacterium* in their colons. These mice showed reduced growth of skin cancer cells and had higher levels of antitumor cytotoxic T lymphocytes (CTL). Species linked to antitumor immune responses were *Bifidobacterium breve*, *B. longum* and *B. adolescentis* [144]. Mice devoid of these species recovered from melanoma and showed an increased in tumour specific CTLs when administered *B. breve* or *B. longum* [144]. Once administered, bifidobacteria proliferate in the nutrient-rich environment created by cell death and necrosis [144]. The specific mechanism by which bifidobacteria or other intestinal bacteria stimulate antitumor immune responses is unknown. They may stimulate the maturation of dendritic cells (DCs) that, similar to antigen-presenting cells (APC), play a role in T-cell activation. *B. longum* BB536 stimulate the development and maturation of interferon γ (IFN-γ) secreting cells. Newborn infants showed an increase in the ratio of IFN-γ/IL-4 secreting T helper (Th) cells (Th1/Th2) when they received *B. longum* BB536 [145].

*Enterococcus faecalis* downregulates the expression of the FIAF (angiopoietin-like protein 4) gene associated with the development of some cancer types [146]. In a mouse model of ulcerative colitis, *E. faecalis* inhibited inflammation by suppressing T helper (Th)-1 and Th17 responses [147]. Heat-killed cells of *E. faecalis* YM-73 enhanced immune modulation by increasing Th1 and reducing Th2-associated cytokines [148].

Gut bacteria could also be used to reduce normal tissue damage during or after radiotherapy. Several studies have shown that probiotic strains of *Lactobacillus* and *Bifidobacterium* reduce radiotherapy side effects [149,150,151,152,153].

LAB are of paramount importance in the prevention of CRC. Insights into the cellular and molecular mechanisms which include apoptosis, antioxidant, immune responses, and epigenetics opened the door for the development of novel therapeutic approaches. Although a wide range of studies has shown the remarkable potential of LAB strains in interfering with colorectal carcinogenesis, conclusive clinical evidence supporting the role of probiotics in CRC treatment is still lacking. More epigenetic studies on LAB are required to demonstrate their effects in cancer prevention. Although several mechanisms of action of LAB in carcinogenesis have been described in in vitro and animal model studies, we are still far from pinpointing the exact cellular signals.

## 3. Bacterial Toxins

Bacterial toxins with medicinal applications have been well studied, especially those produced by *Bacillus thuringiensis* [154], *Clostridium* spp. and *Bacillus* spp. [155]. Botulinum neurotoxin (BoNT), produced by *Clostridium botulinum*, is widely used in ophthalmology, dermatology, and neurology [156]. To date, 108 clinical trials have been registered to evaluate the anticancer properties of bacterial toxins and an additional 98 trials have been registered to study the anticancer properties of immunotoxins (https://www.clinicaltrials.gov/, accessed on 1 June 2022).

### 3.1. Diphtheria Toxin

In 1884, Loeffler injected a pure culture of *Clostridium diphtheriae* into rabbits and pigeons and described the formation of lesions in several organs [157]. Follow-up studies have shown that these lesions were caused by diphtheria toxin (DT), encoded by the *tox* gene located on the genome of corynebacteriophage β [157,158]. The toxin targets elongation factor II (aminoacyl transferase II) and inhibits peptide synthesis [159,160]. The mode of action of DT is summarised in Figure 1.

In many human cancer cells, including hepatocarcinoma, melanoma, and colon, breast, myeloma, prostate, bladder and oral tumours, the *HBEGF* gene, encoding the membrane-anchored precursor of proHB-EGF, is significantly upregulated [161]. Increased levels of proHB-EGF and HB-EGF have been implicated in resistance to chemotherapeutic agents [162]. Cross-reactive material 197 (CRM197), a non-toxic variation of DT, binds to pro-HB-EGF and HB-EGF and inhibits the mitogenic action of HB-EGF by preventing its binding to ErbB (epidermal growth factor) receptors [163]. CRM197 has been used in the treatment of oral squamous cell carcinoma [161]. Tumour formation was completely inhibited in vivo when CRM197 was used in combination with cisplatin, a chemotherapeutic drug [164]. In humans, CRM197 led to an increase in neutrophil and TNF α levels, and a decrease in lymphocyte numbers. The drug, also referred to as BK-UM, proved effective in the treatment of resistant cancers [162,164]. Nam et al. [164] tested the efficacy of intravenously administered BK-UM against the triple-negative breast cancer cell line MDA-MB-231. The size of tumours treated with BK-UM decreased after two weeks. Furthermore, no adverse side effects such as weight loss were reported. In phase I clinical studies, BK-UM was tested against recurrent ovarian and peritoneal cancer cells [162]. Patients were administered four dose levels (1.0, 2.0, 3.3 and 5.0 mg/m^2^), of which 2.0 mg/m^2^ was the most effective. Those treated with 3.3 mg/m^2^ complained of nausea, hypotension, fever, and irritation of the peritoneum [162]. A truncated version of DT, in which the cell receptor-binding domain was replaced by proteins that selectively binds to the surface of cancer cells (referred to as DT385), inhibited angiogenesis and decreased tumour growth in chick chorioallantoic membranes [165]. DT385 also inhibited the growth of Lewis lung carcinoma (LLC) tumours in mice models [165]. Although the results were promising, DT385 entered cells without its R-domain, which triggered non-specific toxicity. To circumvent the problem, immunotoxins that target specific receptors on cancer cells and biochemical processes were developed and approved by the FDA in 1978 [166].

First-generation immunotoxins developed were antibodies randomly linked to non-binding toxins or derivatives thereof [167]. These immunotoxins were non-specific and targeted several receptors on cell surfaces. Second-generation immunotoxins were without cell surface-receptor domains and were much larger, which made penetration of solid tumours more difficult. Both generations of immunotoxins caused severe side effects, such as vascular leak syndrome, haemolytic uremic syndrome and pleuritis [166]. Progress in recombinant DNA technology led to the development of third-generation immunotoxins with less side-effects and improved tumour-penetrating properties. In these immunotoxins the non-specific receptor-binding domain is replaced by the Fv domain of an antibody, either genetically or chemically. Despite being more specific in the targeting of cancer cells, third-generation immunotoxins were not effective in patients vaccinated against diphtheria [168]. Attempts to suppress immune responses with anti-monoclonal antibodies [169] and binding to polyethylene glycol (PEGylation) were unsuccessful [170]. This led to the developing of humanized immunotoxins, i.e., immunotoxins containing a human protein with anti-cancer properties [168]. The first FDA approved humanized immunotoxin, *Denileukin Diftitox* (Ontak, DAB389 IL-2, Eisai Medical Research, Inc., Tokyo, Japan), was constructed by fusing human interleukin-2 (IL-2) with fragment A of DT [171]. IL-2 was genetically fused to the first 388 amino acids of DT, thereby replacing the R-domain [167]. Success rates with Ontak ranged from 30% to 50% [172]. Side effects reported were nausea, diarrhoea and vascular leak syndrome [173]. Despite difficulties encountered in purification of the first recombinant protein expressed by *E. coli* [173], several humanized immunotoxins were developed, all based on DT (Table 1).

### 3.2. Clostridium perfringens Enterotoxin

*Clostridium perfringens* enterotoxin (CPE) is a polypeptide of 319 amino acids [183], arranged in three domains. Domain I represents the binding domain (residues 162 to 309) and domains II and III the cytotoxic domains [184,185]. CPE is produced intracellularly and is only released during endospore germination [183]. Mutations in the TM1 region of CPE (amino acids 81 to 106) resulted in loss of membrane insertion, indicating that it plays a role in pore formation. The mode of action of CPE is illustrated in Figure 2. The amphipathic TM1 region forms a β-hairpin and a β-barrel pore once inserted into the membrane [186]. Amino acids at positions 45 to 53, located upstream of the TM1 region, are responsible for CPE oligomerization [187]. Domains II and III consist of eight β-sheets (two of which span the entire length of the module), two α-helices, and two 3_10_ helical segments [185]. Domain I is a nine-stranded β-sandwich [188]. Briggs et al. [189] reported only two domains. Despite the discrepancy in the number of domains, both studies agreed on the structure of CPE and that the N-terminal domain is divided into two sections.

CPE, encoded by genes α, β, ε and ι located on the genome or on a plasmid, are transcriptionally regulated [190]. The *spo0A* gene, encoding the master sporulation regulator Spo0A, in conjunction with NanR, a transcriptional regulator, and three sporulation-associated sigma factors (SigE, SigK and SigF), are responsible for CPE production and gene regulation [191]. Transcriptional regulation of CPE expression has not been fully elucidated. However, a proposed mechanism of regulation has been compiled from various studies [188,192,193]. An Agr-like quorum-sensing (QS) system phosphorylates Spo0A, which activates the transcription of SigF, SigG, SigK and SigE [190]. SigK and SigE are required for CPE production, as they bind promotors to the *cpe* open reading frame (ORF) [190]. Three promoters (P1-P3) are located upstream of the ORF, with P1 being SigK-dependent, and P2 and P3 SigE-dependent [194].

The β-hairpin loops inserted in the membrane of the target cell create a pore through which cations are channelled. CPE binds to claudins (CLDNs) 3, 4, 5, 6, 7, 8, 9, 14 and 19 [187,188,195,196]. CLDN 4 is localized at tight junctions in normal human prostate epithelial cells (PrECs) but are distributed along the entire surface of cancer PrECs [197]. The size of cancerous tumours treated with CPE was reduced by 59%, suggesting that CLDN 4-targetted CPE treatment may be used to treat prostate cancer. Abedi et al. [198] constructed a recombinant plasmid containing the CPE and prostate stem cell antigen (PSCA) and named it pBudCE4.1-CPE-PSCA. The expression of transgenes introduced into cancer cells, referred to as suicide gene therapy, may be the first step towards developing a vaccine against prostate cancer. The authors [198] reported a 62.6% death rate of PC3 prostate cancer cells. Genes encoding apoptosis were overexpressed, whereas genes encoding cell cycling were repressed. The influx of Ca^2+^ ions activates Ca^2^-dependent proteases and causes cell lysis. Cytochrome C is released from the mitochondrion and caspase 3/7 is activated, leading to apoptosis [13,187,188,199,200,201].

Pahle et al. [202] used suicide gene therapy to treat mice with colorectal cancer. The authors amplified cDNA of CPE by PCR from plasmid pCpG-optCPE to construct a translation-optimized CPE vector (optCPE) and fused the amplicon to genes encoding green fluorescent protein (GFP), resulting in pcDNA3-optCPE-GFP (optCPE-GFP). This construct, and recombinant CPR (recCPE) were used to transfect different cell lines, including CaCo-2 and HT-29, and isogenic Sk-Mel5 and Sk-Mel5 Cldn-3-YFP melanoma cell lines. Colon carcinoma cell lines that overexpressed CLDN 3 and 4 were highly sensitive to recCPE and optCPE, but cells transfected with optCPE displayed rapid cytotoxic effects such as membrane disruption and necrosis. This suggested that suicide gene therapy may be used to suppress colon cancer in cells overexpressing CLDN 3 and 4. Gabig et al. [203] compared the cytotoxicity of CPE against the chemotherapeutics Dasatinib (Das) and Mitomycin C (MMC) used in the treatment of bladder cancer. The cells were killed within one hour when exposed to CPE, compared to 24 h when treated with Das or MMC. Furthermore, after one hour of treatment, 75% of primary bladder cancer cells died (in a 3D culture). Normal cells and cells derived from highly aggressive tumours survived all treatments.

Despite the success of CPE with experimental models, its use in treatment of cancer is limited due to the abundance of CLDNs in normal cells [204]. Shim et al. [200] tested DOX-C-SNP (doxorubicin-loaded C-CPE-polysialic acid) nanoparticles against pancreatic tumour cells in vitro and in vivo and have shown that DOX-C-SNPs accumulated only in tumour cells, without displaying significant cytotoxicity towards non-target cells [200]. A 5.9-fold increase in apoptosis was recorded in orthotopic murine models. Gao et al. [204] reported a decrease in CLDN 4 expression when epithelial ovarian cells (EOCs) were treated with C-CPE and ascribed this to the disruption of TJ proteins. CLDN 4^+^ EOC cell lines were also more sensitive to chemotherapeutic agents, as shown with 59% suppression of tumour growth when cells were treated with a combination of C-CPE and Taxol [204]. Treatment with C-CPE also resulted in the upregulation of genes in the ubiquitin-proteasome pathway that regulates apoptosis and angiogenesis, and downregulated genes involved in metabolic pathways. Becker et al. [205] linked gold nanoparticles (AuNPs) to C-CPE to form a C-CPE-AuNP complex that targets CLDN-overexpressing cancer cells. The Strep-Tag Strep-Tactin fusion system developed by Becker et al. [205] could also be used to conjugate C-CPE to chromophores, thereby allowing imaging and detection of cancer cells. Photonic activation of the AuNPs, referred to as AuNP-mediated laser perforation (GNOME-LP), used in combination with C-CPE is highly specific and targets only CLDN^+^ cells. A 30% and 40% reduction in cell viability was recorded for MCF-7 and OE-33 cells, respectively [205]. C-CPE-targeted GNOME-LP had no significant effect on the survival of cells in the control group. Gabig et al. [203] have shown that C-CPE treatment of RT4 (non-invasive superficial) cancer cells enhanced the toxicity of Das and MMC. Moreover, a drastic decrease in CLDN 4 expression was recorded, without affecting normal cells [203]. Nanoparticles loaded with fluorescent rhodamine dye and superparamagnetic iron oxide, linked to C-CPE (CPE_290–319_) were used to target CLDN 3 and CLDN 4 in cancer cells [206]. This technique may be used to determine the aggressiveness of cancer tumours.

### 3.3. Botulinum Toxins

Botulinum toxins (BoNT), produced by *C. botulinum*, *Clostridium butyrricum*, *Clostridium barati*, and *Clostridium argentinensis* are used in the treatment of muscle disorders [207], anismus [208], tremors [209], dystonia [210], cancer [211], and severe pain [212]. The inactive single-chain polypeptide is nicked by a protease to form a di-peptide of 100 kDa and 50 kDa [207]. The light chain (LC) is located at the N-terminal and contains the catalytic domain (C-domain), whereas the heavy chain (HC) is divided into the central translocation (T) domain and the C-terminal receptor-binding (R) domain [213]. The toxin associates with non-toxic neurotoxin-associated proteins (NAPs) to form a 300 to 900 kDa protoxin resistant to stomach acid and improved ability to be translocated across the intestinal epithelial barrier [207]. Eight types of botulinum toxins (A, B, C1, C2, D, E, F, and G) and a novel serotype (BoNT/H), isolated from an infant with botulism, have been described [207,214]. The mode of action of BoNT is illustrated in Figure 3.

BoNT binds to ecto-acceptors (polysialogangliosides) on the presynaptic cell surface of cholinergic neurons and is internalized via synaptic vesicles (SVs) or membrane-trafficking proteins (synaptotagmins). Acidification of the SVs leads to the activation of acetylcholine (Ach) transporter proteins that import Ach, the LC and HC domains of BoNT. Heat shock protein 90 (Hsp90) and thioredoxin reductase-thioredoxin (TrxR-Trx) cleaves the LC and liberates it into the cytosol. The C-domain is a Zn^2+^-dependent endopeptidase that cleaves proteins in the soluble SNARE protein complex. This complex is responsible for exocytosis and the fusing of Ach-containing vesicles with the plasma membrane, allowing the release of Ach. For more information BoNT mode of action, the reader is referred Nigam and Nigam [156], Gul et al. [213], Dolly et al. [215], Choudhury et al. [216] and Huang et al. [217].

Huang et al. [217] performed one of the earliest studies using BoNT/A to treat HIT-T15 insulinoma cells. Although the authors did not determine whether BoNT/A can be used to kill cancer cells, they showed that a transient transfection with the toxin can inhibit insulin expression. This paved the way for more in vivo and in vitro studies using BoNT. Treatment with BoNT/A render cancer cells radiosensitive [218] and may be used in the treatment breast and prostate cancer [211,219]. BoNT may also be used as an immunotoxin [220]. Toxin A induces apoptosis, inhibits the proliferation of LNCaP (infraclinical prostate cancer) cells, as shown with in vitro and in vivo studies [219]. In a separate study [221], toxin A showed cytotoxicity against cell lines LNCaP and PC-3 (prostate cancer), most probably due to the phosphorylation of phospholipase A2. Cell death of T47D breast cancer cells was attributed to the induction of caspase 3- and 7-dependent apoptosis [222]. Toxin C induced apoptosis and cell death in differentiated human neuroblastoma cells (SH-SY5Y and SiMa) [223]. Other cell-line anticancer studies performed with BoNT are listed in Table 2.

BoNT inhibits the release of neurotransmitters [207] and may be used as a painkiller in cancer treatment. Van Daele et al. [6] were the first to report on the analgesic effect of BoNT/A. The authors treated patients with painful spams of the sternocleidomastoid muscle. Injection with BoNT/A relieved the pain in four of six patients. Wang et al. [228] administered BoNT/A to a lung cancer patient with Raynaud phenomenon and previously treated with chemotherapy. Conventional treatment of neoplasms is generally ineffective. However, after BoNT/A treatment patients reported relief in symptoms, with no adverse side effects. Incobotulinumtoxin A (INCO), a BoNT/A preparation used in clinical settings, has also been used in the treatment of cancer-related pain [12]. Twelve patients with head, neck and breast cancer were enrolled in the study. Two patients passed away due to advanced cancer, one developed a skin rash, and another did not return due to poor general health. Three of the remaining eight patients reported an improvement in their quality of life. Pain amelioration was assessed using the Visual Analog Scale (VAS). All eight patients reported a significant improvement. A significant satisfaction of treatment was reported by seven of the eight patients by self-assessment using the Patients’ Global Impression of Change Scale. Dana et al. [10] used Botox and Dysport to test the efficacy of BoNT/A in pain relief of neck and head cancer patients suffering from radiotherapy-induced trismus and masticator spasms. One month after BoNT/A injection, a significant improvement in pain and spams were recorded, but no improvement in trismus. No adverse side effects were observed and the authors concluded that BoNT/A may be prescribed to patients with radiotherapy-induced pain or muscle spasms. De Groef et al. [229] studied the effect of a single BoNT/A injection in conjunction with physical therapy on breast cancer patients that underwent a mastectomy. Of the 50 patients, 25 received the injection (intervention) and 25 a placebo. After three months of treatment, a significant change in pain was observed in the upper limb of patients from the intervention group. Other studies using BoNT/A in the treatment of cancer-related or cancer therapy-related pain that were successful included post-radiosurgical neck contracture [230], frontotemporal glioblastoma related pain [231], and postoperative pain in patients that underwent a mastectomy and tissue expander reconstruction [232].

### 3.4. Pseudomonas aeruginosa Toxin

*Pseudomonas aeruginosa* produces a potent 66-kDa A-B toxin (PE, also known as exotoxin A or ETA) that inhibits protein translation [233,234]. The A domain has enzymatic activity, and the B domain acts as a cell-binding moiety [235,236]. The first 25 amino acids form a highly hydrophobic signal sequence that is removed during secretion [237]. The remaining 613 amino acids is divided into three domains. Domain Ia is the receptor binding domain (first 252 amino acids) that attaches to target cells, domain II (amino acids 253–364) facilitates the translocation of PE across the cell membrane, and domain Ib (amino acids aa 365–404) together with domain III (amino acids 405–613), represents the catalytic part of toxin PE [238]. When secreted, the terminal amino acid residue of PE (lysine 613) is cleaved by a host plasma carboxypeptidase, which converts the REDLK (amino acids 609–613) motif into REDL (amino acids 609–612) [239]. PE interacts with the low-density lipoprotein receptor-related protein 1 (LRP-1) via the Ia domain and is then internalized by endocytosis. In the early endosome, which is acidic, PE dissociates from the receptor and changes conformation, exposing the furin-cleavable motif to be cleaved by furin into two fragments of approximately 28 kDa (279 amino acids) and 37 kDa (333 amino acids) [240]. The 28 kDa fragment consists of domain Ia and parts of domain II. The 37 kDa fragment contains parts of domain II, domains Ib, and domain III and has enzymatic activity. The 37-kDa fragment migrates from the late endosome to the trans-Golgi network (TGN) and from there to the ER via the retrograde pathway. The C-terminal REDL motif interact with KDEL receptors on the TGN [241]. The mode of action of PE is illustrated in Figure 4.

Moxetumomab pasudotox [241] is a recombinant anti-CD22 immunotoxin consisting of a single chain antibody fragment of a mouse anti-CD22 monoclonal antibody (scFv) fused to a *Pseudomonas* endotoxin A (PE38 domain). The immunotoxin binds to CD22 antigen expressed on B cells in various hematological malignancies. Upon internalization and intracellular proteolysis, the cytotoxic fragment (PE38) is released, which then induces cell death by apoptosis [241,242]. In September 2018, Moxetumomab pasudotox (LUMOXITI™; AstraZeneca, Cambridge, UK) was approved by the U.S. Food and Drug Administration (FDA) for the treatment of relapsed or refractory hairy cell leukemia [243]. Immunotoxins containing PE are highly immunogenic in patients with normal immune systems, but less so in patients with hematologic malignancies, whose immune systems are often compromised. SS1P, a first-generation, mesothelin-targeted immunotoxin, demonstrated little activity as a single agent [244]. In patients with solid tumors, neutralizing antidrug antibodies (ADAs) directed against PE developed after only three infusions of SS1P. To delay the development of high-titer ADAs, SS1P was combined with a preconditioning regimen of lymphocyte-depleting chemotherapy [245]. LMB-100, a second-generation recombinant immunotoxin that targets the glycoprotein mesothelin on the surface of cancer cells is composed of a humanized antimesothelin antibody fragment fused to a truncated PE A [246]. The maximum tolerated dose (MTD) of LMB-100 was 140 µg/kg, administered every other day over 3 weeks [246]. Although LMB-100 was less immunogenic than SS1P, most patients developed antidrug antibodies after 2 cycles. Phase 2 clinical studies with LMB-100 plus pembrolizumab is conducted on patients with mesothelioma and lung cancer [246]. Several studies are devoted towards the developing of PE-based recombinant immunotoxins (RITs), especially against mesothelin and other proteins on solid tumors. For more information, the reader is referred to the review by Mazor and Pastan [247].

## 4. Antibiotics

Although antibiotics are mainly used as bactericidal or bacteriostatic agents, some display anticancer properties and are classified as anthracyclines, of which Actinomycin D (Dactinomycin), Bleomycin, Doxorubicin (adriamycin and doxil), Epirubicin (ellence), Cerubine and Daunorubicin (DaunoXome), Novantrone (mitoxantrone), Mitomycin C, Spergualin and Epothilone are the best known.

### 4.1. Actinomycin D

Actinomycin D, produced by *Streptomyces antibioticus* and *Streptomyces parvulus* has two pentapeptide lactone rings and a 2-aminophenoxazine-based chromophore [15]. Intercalation of the chromophore into DNA inhibits transcription and prevents the growth of tumour cells [248]. Interaction of actinomycin D with DNA is facilitated by a GpC (guanosine-cytosine) base pair. Increased binding was obtained with the formation of hydrogen bonds between L-threonine residues in the pentapeptide rings and the amino-terminal of guanosine [249]. Other amino acids of the two lactone rings, including proline, *N*-methylglycine and methylvaline, facilitates the binding of actinomycin D to the minor groove of DNA, thereby improving the stability of the interaction [15,249]. Despite the toxic effects of actinomycin D, such as tissue necrosis, dermatoxicity and gastrointestinal enterotoxicity, the drug has been approved for treatment of Wilms’ tumour, gestational choriocarcinoma, neuroblastoma, germ cell cancers, trophoblastic tumours, Ewing sarcoma and rhabdomyosarcoma [250,251,252,253]. Actinomycin D has also been used in combination with antitumor agents to treat high-risk malignancies [249]. Actinomycin D increased the therapeutic efficacy of antitumor agents such as RG7787, a mesothelin-targeting immunotoxin. RG7787 (also referred to as LMB-100) is a recombinant immunotoxin formed through conjugation between exotoxin A of *P. aeruginosa* and anti-mesothelin Fab [246]. Synergistic cytotoxicity of actinomycin D and LMB-100 towards mouse xenografts of pancreatic and stomach cancer cells were illustrated with Phase I clinical trials [254]. This combination led to apoptosis via an extrinsic pathway and resulted in noteworthy regression of the tumours. Anticancer drugs containing Actinomycin D are available in the market under the trade names Cosmegen and Lyovac [8].

### 4.2. Bleomycin

Bleomycin is produced by *Streptomyces verticillus* [255,256]. A combination of Bleomycin A2 (C_55_H_84_N_17_O_21_S_3_, Mw. 1415.56 Da) and Bleomycin B2 (C_55_H_84_N_20_O_21_S_2_, Mw. 1425.52 Da) was approved by the FDA in July 1973 [207]. The mode of action is described as a two-step process. In the first step Bleomycin chelates metal ions (primarily iron) and produces a pseudoenzyme. The second step is the enzymatic conversion of oxygen to superoxide and hydroxyl free radicals damaging DNA [257,258]. Bleomycin is used to treat Hodgkin’s lymphoma, non-Hodgkin’s lymphoma, testicular cancer, ovarian cancer, and cervical cancer [259,260], and is commercially available as Bleomycin USP and Blenoxane [8].

### 4.3. Doxorubicin, Epirubicin, Daunorubicin and Novantrone

Doxorubicin (adriamycin and doxil; C_27_H_29_NO_11_), produced by *Streptomyces peucetius* var. *caesius* [15,27,261], has amphipathic properties owing to a water-insoluble aglycone (adriamycinone, C_21_H_18_O_9_) and water-soluble amino sugar group (daunosamine, C_6_H_13_NO_3_). Its anti-cancer properties was first reported in 1969 [8] and has been approved by the FDA for treatment of malignant lymphoma, soft tissue sarcoma, and breast-, liver-, ovary-, neck-, head-, gastric- and childhood cancers [15,213]. Doxorubicin binds to DNA and RNA polymerases, which prevents DNA replication and transcription [261], intercalates with DNA and removes histones from chromatin during transcription [8]. The formation of covalent complexes between topoisomerase-II and DNA [15] leads to single- and double-strand DNA breaks and apoptosis [261]. Doxorubicin also binds to cardiolipin and mitochondrial creatine kinase [8]. Patients treated with doxorubicin displayed adverse side effects such as fatal cardiotoxicity and nonspecific cytotoxicity [15]. A nano-drug delivery system, based on liposome-encapsulation of doxorubicin (e.g., Doxil^®^), has been approved for treatment of ovarian, breast- and AIDS-related Kaposi’s sarcoma. Doxorubicin and mitomycin C encapsulated in polymer-lipid hybrid nanoparticles (PLNs) were 20 to 30 times more active than the drugs in separate form and killed multidrug-resistant human breast cells MDA435/LCC6 more effectively [262]. Examples of drugs containing doxorubicin are Myocet, Doxorubicin-Ebewe, Adriblastine PFS, Caelyx, Doxorubicin medac, and Doxorubicinum Accord [8].

Epirubicin (ellence) is a semisynthetic derivative of doxorubicin, with a hydroxyl group in the 4′ position of the daunosamine ring [263]. It is mainly used in the treatment of early or metastatic breast cancer, but also other tumours such as lung, bladder, gastric, ovarian and hepatocellular carcinoma, and lymphatic cancers [264]. The mode of activity of Epirubicin is similar to that of Doxorubicin and also inhibits topoisomerase II activity [265].

Daunorubicin or Daunomycin, also referred to as Cerubine, is produced by *Streptomyces peucetius* and differs from doxorubicin by lacking a hydroxyl group at the the 14th position [266]. The name Daunomycin is derived from the pigment aglycone daunomycinone and the amino sugar daunosamine [267]. The liposome-encapsulated form of daunorubicin, referred to as DaunoXome, is more stable in an aqueous solution and is more toxic towards certain types of solid tumours [266].

Novantrone (mitoxantrone) is chemically related to doxorubicin and acts as a potent immunosuppressive agent for treatment of multiple sclerosis [268]. Novantrone inhibits the proliferation of B and T lymphocytes as well as macrophages, kills antigen-presenting cells, and suppresses the migration of activated leukocytes [269]. Other modes of action for mitoxantrone include lowering the secretion of IFN-γ, TNF-α, and IL-2 [270].

### 4.4. Mitomycin C, Duramycin and Epothilones

Mitomycin C (C_15_H_18_N_4_O_5_, Mw. 334 Da), produced by *Streptomyces caespitosus* [271], is an aziridine [(CH₂)₂NH]-containing antibiotic that cross-links DNA and inhibits alkylation [272]. The drug is used in the treatment of bladder, colorectal and pancreatic cancers, head and neck sarcoma, and lung-, hepatic and esophageal carcinoma [273]. Spergualin (C_17_H_37_N_7_O_4_, Mw. 403.53 Da), produced by *Brevibacillus laterosporus* BMG162-aF2, repressed fibrosarcoma cells (M5076) and cell lines of rat hepatomas (AH66, AH66F), as well as leukemia in mice models [274,275]. Duramycin induces apoptosis and reduces the proliferation in tumour cells [276,277], a phenomenon that may be ascribed to its high affinity for phosphatidylethanolamine [278]. The cytotoxicity of duramycin was reduced by fusion to IgG [279,280]. This did not influence binding to phosphatidylethanolamine. Fusion to IgG guide host immune cells to apoptotic cells, resulting in enhanced phagocytosis. Tumour growth in mice was inhibited after treatment with duramycin-IgG [280]. Since duramycin binds to PE and the Fc region (fused) IgG antibodies, it interacts with phagocytic cells to enhance phagocytosis. Duramycin is cleared from the site effectively soon after inducing apoptosis in cancer cells, via phagocytosis, which would explain its lower cytotoxicity to surrounding normal cells. Epothilone A and B, produced by *Sorangium cellulosum*, is classified as a macrolide polyketide [15]. Both variants inhibit mitosis and induce the formation of α/β-tubulin polypeptide heterodimers [186]. Low dosages of epothilone inhibits cell growth without blocking mitosis [186]. Ixabepilone (IXEMPRA), a synthetic analogue of epothilone B, represses the growth of a variety cancer cells. In 2007, the FDA approved IXEMPRA for treatment of breast cancer cells resistant to treatment with taxanes such as paclitaxel and docetaxel, anthracycline and capecitabine. IXEMPRA combined with capecitabine was more effective than capecitabine in the treatment of breast cancer cells resistant to taxane and anthracycline.

## 5. Bacteriocins

Bacteriocins of Gram-negative bacteria are divided into four main classes: colicins, colicin-like, phage-tail-like bacteriocins, and microcins [281]. Colicins are protease-sensitive, heat-sensitive and has a high-molecular weight (30–80 kDa). Most *E. coli* strains have genes encoding colicins. These proteins are expressed when cells experience stress and usually leads to self-destruction due to co-expression with lysis protein [282]. Depending on the mechanism of interaction with the target cell, colicins are divided into three main groups, i.e., pore-forming, nuclease, and peptidoglycan degrading. Bacteriocins of Gram-positive bacteria are ribosomally-synthesized peptides with bacteriostatic or bacteriolytic activity and are usually below 10 kDa in size. These antimicrobial peptides are grouped into four classes. Class I are linear peptides, produced as pre-peptides, and are converted to active, mature, peptides after post-translational modification. They contain several disulphide bridges and unusual polycyclic thioether amino acids such as dehydrobutyrine, dehydroalanine and lanthionine. Class II bacteriocins do not contain lanthionine and do not undergo post-translation modification, except for the removal of the leader peptide [16]. Bacteriocins from both classes are thermostable. Class III bacteriocins are thermolabile and have a molecular mass exceeding 10 kDa. Class IV bacteriocins are circular [9,283].

The first report of a bacteriocin displaying anticancer properties was published in the late 1970s, but this was a crude extract [284]. Since then, several reports of bacteriocins with anticancer properties (also from our own group, unpublished), have been reported [285]. To the best of our knowledge only three bacteriocins have been patented for their anticancer properties, i.e., plantaricin A produced by *L.*
*plantarum* C11 [8,286,287], microcin E492 produced by *Klebsiella pneumoniae* [288,289,290,291] and Pep27anal2, a 27-amino acid peptide produced by *Streptococcus pneumoniae* [291,292]. Anticancer properties have also been reported for bovicin HC5 produced by *Streptococcus bovus* HC5 [293,294], colicins A, E1, E3, and U isolated from *E. coli* [295,296,297,298,299,300], pyocins from *P.*
*aeruginosa* [301], nisin from *Lactococus lactis* [302], and pediocins from *Pediococcus* and other lactic acid bacteria [303,304].

Pyocin S2, produced by *P. aeruginosa* M47 (PAO 3047), inhibited the growth of cancer cells XC, TSV-5, mKS-A TU-7, HeLa-S3, and AS-II, but did not affect normal cells such as mice cells BALB/3T3, rat kidney cells and human lung cells [305]. Abdi-Ali et al. [306] reported cytotoxic activity of pyocin S2 towards HepG2 cells and human immunoglobulin secreting (Im9) cells isolated from multiple myeloma [306]. Pep27anal2, an analog of signal peptide Pep27, produced by *S. pneumoniae*, causes caspase-dependent and cytochrome C-independent apoptosis in cancer cells, as shown with studies on OCI-AML-2 and HL60 (leukemia) cells, Jurkat cells, and MCF-7 and SNU-601 (adenocarcinoma) cell lines [292]. Bovicin (2.4 kDa), produced by *Streptococcus bovis*, showed cytotoxicity towards human breast cancer (MCF-7) and human liver hepatocellular carcinoma (HepG2) cell lines in a concentration-dependent manner [294]. Colicins, produced by *E. coli*, are larger than the average bacteriocins (>20 kDa). Colicin E3 showed cytotoxic and cytocidal effects against HeLa (human cervical cancer) cells and cleaves rRNA [297]. Colicin produced by *E. coli* HSC10 degraded DNA and is cytotoxic towards mammalian cells [295]. Colicin from the same strain, referred to as verotoxin 1, acted anticarcinogenic against human ovarian cell lines but protected cells in a murine metastatic fibrosarcoma model [296]. Colicins E1-E5 displayed cytotoxicity towards hamster V79 fibroblast cells [297]. Colicin A, E1, E3 and U caused cell cycle arrest in human fibroblast cell lines MRC5, MCF-7, MDA-MB-231 (mammary tumor), HOS (bone osteosarcoma), and HS913T (fibrosarcoma) [298]. In general, colicins depolarize the cytoplasmic membrane, prevents the synthesis of peptidoglycan, degrade DNA and RNA, seize cell cycles, and causes necrosis [299].

The interaction of bacteriocins with cancer cells is ill researched. Kaur and Kaur [16] hypothesized that cancer cells increase the expression of negatively charged cell-surface molecules when exposed to bacteriocins and, by doing so, promote cytotoxicity and apoptotic cell death. Azurin, an anticancer copper-containing bacteriocin of 14 kDa (Figure 4) produced by *P.*
*aeruginosa* [307], provides some indication as to how bacteriocins may enter cancer cells. The peptide enters human cancer cells through receptor-mediated endocytosis [308] or, as reported with studies on breast cancer cells lines MCF-7, ZR-75-1 and T47D, via caveolin-mediated pathways [309]. The p28 domain of azurin (Figure 5) facilitates cell crossing and promotes apoptosis [310]. Sections within p28 have different adhesion and cell penetration properties, as summarized in the legend of Figure 5. Once inside a cancer cell, azurin attaches to the DNA-binding domain (DBD) of the tumor-suppressor protein p53 (Figure 6) and increases the intracellular level of the protein by inhibiting the binding of the E3 ubiquitin ligase COP1 to p53 [311]. This leads to repression of cell growth and apoptosis [312]. Amino acids 88 to 113 in the C-terminal of azurin repressed the growth of MCF-7 breast- and DU145 prostate cancer cells [313]. More recent studies have shown that azurin also interferes with non-receptor tyrosine kinase (NRTK) signalling pathways [314]. Increased intracellular levels of p53 and Bax protein (a central cell death regulator) were detected in cells exposed to azurin [314]. This led to the release of cytochrome C in the cytoplasm, and activation of caspases 9 and 7 [8]. Azurin also reduces VEGFR-2 tyrosine kinase activity, thereby preventing the formation of new blood vessels and the expression of P-cadherin, a glycoprotein maintaining the structural integrity in epithelial tissue [314]. Phase I clinical trials were performed with p28 on recurrent, difficult to treat, and progressive solid tumours in adults, and on tumours from the central nervous system (CNS) of paediatric patients. No significant immune response or adverse side effects were reported [315]. Protein p21, a short amino acid fragment of azurin overexpressed in MCF-7 cells, inhibited cyclin-dependent kinases and prevented cell growth [316].

A few bacteriocins produced by LAB are worth pointing out and are summarized here. Plantaricin A (2.98 kDa), a peptide pheromone produced by *L. plantarum* C11, binds to negatively charged phospholipids and glycosylated proteins in cell membranes of cancerous and normal cells, leading to the destabilization of cytoplasmic membranes [318]. The peptide also induces apoptosis and necrosis in Jurkat cells, accompanied by an increase in caspase 3 levels [286,287]. Microcin E492 (7.8 kDa) causes cell shrinkage, DNA fragmentation, release of phosphatidylserine and calcium ions, and apoptosis of cancerous cells such as HeLa, Jurkat. RJ2.25 and colorectal carcinoma cells [289]. Normal bone marrow cells, splenocytes, KG-1, human tonsil cells, and nontumor macrophage-derived cells were not affected [289].

An in vitro study showed that lacticin A164 and BH5, produced by *L.*
*lactis* subsp*. lactis*, inhibited the growth of *Helicobacter pylori* [319], which is responsible for a sizeable number of gastrointestinal cancers. Probiotic preparations with these strains may thus be used to control of *H. pylori* infection and related intestinal cancers. Nisin, also produced by *L. lactis* subspecies *lactis,* has been used in the food industry for over 50 years [16,27]. Joo et al. [320] reported a decrease in the proliferation of head and neck squamous cell carcinoma (HNSCC) cells when exposed to nisin and illustrated decreased tumour growth in an oral cancer floor-of-mouth mouse model [320]. Athymic nude mice (NCr-nu/nu strain) were used. Gene array analyses performed on HSC-3 oral squamous cell carcinoma (SCC) cells treated with nisin revealed an increase in genes regulating calcium transport, membrane lipid functioning and apoptosis. The gene most upregulated encodes the cation transport regulator CHAC1 (ChaC Glutathione Specific Gamma-Glutamylcyclotransferase 1), which is known to be activated by oxidized phospholipids [320]. Since nisin interacts with phospholipids in cell membranes, especially phosphatidylcholine, it is possible that CHAC1 may be a “downstream” nisin target. In in vitro experiments conducted with nisin ZP on HNSCC cells and with NCr-nu/nu mice much higher levels of apoptosis were reported compared to treatment with nisin [321]. Apoptosis of HNSCC cells increased with higher concentrations of nisin ZP [321]. This coincided with a decrease in cell proliferation, clonogenic capacity, and sphere formation [321]. Nisin ZP induced apoptosis through a calpain-dependent pathway in HNSCC cells but not in human oral keratinocytes [321]. Apoptosis of human umbilical vein endothelial cells (HUVEC) induced by nisin ZP coincided with a decrease in vascular sprout formation (in vitro*)* and lowering of intratumoral microvessel density (in vivo) [321]. No abnormalities in organ tissue, inflammation, fibrosis or signs of necrosis were observed in mice treated [321], suggesting that nisin ZP may be a promising alternative in cancer therapy. Reports of nisin showing potential in the treatment of colorectal cancers [321,322,323] and growth inhibition of blood, breast, brain, colon, gastrointestinal, liver and skin cancer cells [324] have been published.

Preet et al. [325] studied the synergistic effect of nisin and doxorubicin on dimethylbenzanthracene-induced skin cancer in murine models. Doxorubicin-alone-treatment and nisin-alone-treatment resulted in a mean tumour burden reduction of 51.3% and 14.18%, respectively. Mice treated with a combination of nisin and doxorubicin displayed a reduction of 66.82% in the mean tumour burden. Nisin combined with 5-FU killed skin cancer cells, induced with 7,12-dimethylbenz(a)anthracene, in vivo [326]. Anticarcinogenic properties was ascribed to modulation of apoptotic, angiogenic and cell proliferative pathways.

Pediocin PA-1, produced by *Pediococcus acidilactici* K2a2-3, is cytotoxic towards human colon adenocarcinoma (HT29) and HeLa cells [303]. Pediocin CP-2, produced by *P. acidilactici* MTCC 5101 and rec-pediocin CP-2 (recombinant) cells are cytotoxic towards MCF-7, HepG2, HeLa and mouse spleen lymphoblast (Sp2/O-Ag14) cells and induces apoptosis [304].

Bacteriocins may thus play a key role in prevention of intestinal and skin cancers. Only a few of such studies have been published and most were performed in vitro. Should these results be confirmed by in vivo studies, methods will have to be developed to protect these peptides from degradation by gastrointestinal enzymes and enhance their activity. Bacteriocin-producing strains with anticarcinogenic properties may be included in probiotic supplements. We are, however, still a long way from understanding the efficacy of bacteriocins in anticancer therapy and whether these peptides should be used alone or in combination with chemotherapeutic agents.

## 6. Non-Ribosomal Peptides and Polyketides

Non-ribosomal peptide synthetases (NRPS) were discovered in 1968 by Gevers and co-workers when they studied gramicidin S production by *Bacillus brevis* [327]. By adding RNAse and puromycin (a ribosome inhibitor) to *B. brevis* extracts containing gramicidin S, Gevers et al. [328] observed that aminoacyl-tRNA synthetases and tRNA were not used in the production of gramicidin S. Subsequent studies have shown that non-ribosomal peptides are produced by enzyme complexes in a nucleic acid-independent manner [329]. Further research is needed to determine if non-ribosomal peptides can be used in cancer therapy [8]. Polyketides are produced non-ribosomally by type 1 polyketide synthetases (PKSs) [330]. Almost one-third of all pharmaceuticals are polyketides which can be attributed to their structural and biological diversity. Type 1 PKSs are modular enzyme assemblies and act successively to elongate the polyketide chain. Different domains can be found in PKSs such as ketoreductase, dehydratase, and enoylreductase with conserved domains such as acyl carrier protein, ketosynthase and acyltransferase. Readers are referred to Baindara and Mandal [8] for further information non-ribosomal peptides and polyketides. Noteworthy is that no non-ribosomal peptides and polyketides with anticancer properties have been reported for lactic acid bacteria.

## 7. Phenylpropanoids

Phenylpropanoid-derived metabolites from plants inhibits the growth of several cancer cell types [331]. However, Bacteroides thetaiotaomicron, Bacteroides eggerthii, Bacteroides ovatus, Bacteroides fragilis, Parabacteroides distasonis, Eubacterium hallii and Clostridium bartlettii ferment phenylalanine, tyrosine and tryptophan to phenylacetic acid (PAA) and 4-hydroxylphenylacetic acid (4-hydroxyPAA) in the colon [332]. An increase in cell numbers of Bacteroides fragilis was observed in patients with advanced stages III and IV) CRC [333].

The phenylpropanoid verbascoside protects the GIT from oxidative stress and represses the growth of MKN45 gastric epithelial cancer cells, but also stimulates appetite [334,335]. Further research on the stability of phenylpropanoids is required. Acteoside, a verbascoside, has anti-inflammatory properties [336] and prevents the onset of chronic diseases, including cancer [337,338], but is degraded by gut microbiota [339]. Further research is required to understand the degradation of phenylpropanoid by gut microbiota, and exactly how these compounds modulate inflammatory and microbial processes.

## 8. Phenylflavonoids

Xanthohumol (XN), a prenylated flavonoid found in hops, has promising anticancer properties [340]. Gut microbiota metabolites XN to 8-prenylnaringenin (8-PN), a very potent phytoestrogen with anticancer activity, as demonstrated with SK-MEL-28 and BLM human metastatic melanoma cells [341] and MCF7 breast cancer cells [342]. 8-PN also inhibited cell proliferation of the HT-115 and HT-116 colon cancer cells [342,343].

## 9. Natural Purine Nucleosides

Adenosine and inosine (formed by the catabolism of adenosine) bind to adenosine receptors, and initiates cAMP production and the phosphorylation of kinase-1 and -2 [344]. Inosine also enhances T cell antitumour activity in colorectal, bladder, and melanoma cancer cells [345]. Studies on bladder cancer cells indicated that inosine enhanced the function of anti-CTLA-4, causing to increase infiltration of IFN-γ+ CD4+ and IFN-γ+ CD8+ T-cells into the tumour, as well as reducing overall tumour weight [345]. CTLA-4, also known as CD152, is a protein receptor that functions as an immune checkpoint downregulating immune responses, a process referred to as immune checkpoint blockade (ICB). Inosine produced by *Bifidobacterium pseudolongum* increased the activation of a cDC-dependent TH 1 cell circuit and led to the enhancing of ICB in murine models with intestinal and epithelial tumours [345].

## 10. Short Chain Fatty Acids

Diet plays an important role in CRC and may promote the formation of tumours [346]. SCFAs such as butyrate, acetate, propionate and lactate are largely produced in the colon by *Bifodobacterium*, *Lactobacillus*, *Lachnospiraceae*, *Blautia*, *Coprococcus*, *Roseburia*, *Faecalibacterium*, *Clostridium*, *Eubacterium*, and species converting lactate and acetate, e.g., *Anaerostipes* spp. [347,348]. These SCFAs adhere to free fatty acid receptors (FFARs), e.g., GPR43 (FFAR2) and GPR41 (FFAR3) located on the surface of IECs [349]. Patients diagnosed with CRC had less *Bifidobacterium* spp. and lower levels of SCFAs [350]. Butyrate also protects intestinal barrier function by up-regulating the tight junction protein claudin-1 [351,352,353]. Other functions of butyrate include maintaining a balanced state of oxygen [301] and suppressing inflammatory responses [354,355]. The latter is achieved by downregulating histone deacetylase (also referred to as lysine deacetylase) inhibitors (HDACi) [354,355]. An increase in de-acetylated histones (due to the inhibition of HDACi), together with a decline in gene transcriptions, leads to autophagic cell death, the activation of extrinsic and/or intrinsic apoptotic pathways, an increase in production of reactive oxygen species (ROS), and a decrease in the expression of pattern recognition receptors, kinases, transcription regulators, cytokines, chemokines, and growth factors [356,357]. Molecules released from dying cells are recognised by nucleotide-binding oligomerization domain (NOD)-like receptors (NLRs) and form specific protein-protein interactions. These interactions, also prevalent in lymphocytes, macrophages and dendritic cells, play a key role in the regulation of cytokines, chemokines and the expression of genes coding for the production of antimicrobial compounds, collectively referred to as the innate immune response [358,359,360]. Downregulation of NLRs prevents the formation of multi-protein inflammasomes, the signaling of caspase-independent nuclear factor kappa B (NF-κB) and mitogen-activated protein kinase (MAPK). These cascades of events counteract autoimmune and inflammatory disorders and as recently shown, represses the growth of cancerous cells [361].

Butyrate and propionate are more effective than acetate in inhibiting the growth of HT29 cells [362]. Butyrate significantly increased apoptosis in cancer cells [363] and activates ornithine decarboxylase, resulting in the inhibition of polyamine metabolism and an increase in alkaline phosphatase activity [364]. Butyrate also activates GPR41/GPR43 receptor signaling pathways [364], thus preventing the proliferation of cancer cells [365]. GPR43 is predominantly present in the large intestine of healthy cells, but less so in colon cancer cells [366]. Manipulation of HCT8 human colonic adenocarcinoma cells to express the GPR43 receptor led to an increase in apoptosis and G0/G1 cell cycle arrest [366]. Concluded from these studies, the GPR43 receptor serves as a tumour suppressor. SCFAs were ineffective against HCT-116 colon cancer cells with a deletion in the *p21* gene, suggesting that p21, a cell cycle inhibitor and anti-proliferative effector, is important in repressing tumour growth. Another study has shown that *p21* is regulated by the p53 transcription factor [367]. Diets rich in SCFAs suppresses T cell-mediated autoimmune responses, most probably through regulation of cytokine expression [364]. The ability of butyrate to de-repress epigenetically silenced genes in cancer cells, such as cell cycle inhibitor p21 and the pro-apoptotic protein Bcl-2, has important implications for cancer prevention and therapy. Lightfoot et al. [368] tested possible epigenetic modifications induced by LTA-deficient *L. acidophilus* and found that oral NCK 2025 enhances the expression of tumor suppressor genes [368,369]. This indicates that differential epigenetic regulation of CRC-related genes by NCK2025 represents a potential therapy against CRC.

Overall, SCFAs are promising specifically in the context of colon cancer. Future studies should evaluate the effect of SCFAs on other cancer types, e.g., pancreatic and gastric cancer. Studies should also explore the impact of SCFAs on the efficacy and safety of standard chemotherapy and the prognosis of cancer.

## 11. Enzymes

Four enzymes with anticancer properties have been reported, i.e., arginine deiminase (ADI), produced by *Mycoplasma hominis* and *Mycoplasma arginine*, asparaginase (ASNase), produced by *E. coli* and *Erwinia chrysanthemi*, glutaminase [370] and methionase [371]. ADI converts arginine to citrulline and ammonia in vivo [372]. Pegilation of ADI (ADI-PEG20) significantly increased the half-life of ADI in serum and decreased its antigenicity [373], rendering ADI much more effective against cancerous cells. The mode of activity of ADI-PEG20 is ascribed to loss of argininosuccinate synthetase (ASS) activity and, thus, the inability to synthesize arginine from citrulline [374]. Due to this, growth of hepatocellular carcinoma cells (HCC), auxotrophic to arginine, were repressed, with evidence of apoptosis [375]. Treatment of metastatic hepatocellular carcinoma cells with ADI-PEG20 entered phase II clinical trials [376]. Autophagy was induced in prostate cancer cells (CWR22Rv1) treated with ADI-PEG20 [374]. ASNase degrades asparagine, which results in a severe reduction in protein synthesis and thus growth inhibition of cancer cells [377,378]. Studies performed with human cells lines (pediatric medulloblastoma, DAOY, and glioblastomas GBM-ES and U87) indicated that inhibition of cell growth by ASNase is dose-dependent [379]. ASNase has been used in the treatment of acute lymphoblastic leukemia (ALL), acute myeloid leukemia, ovarian carcinoma, myelosarcoma, Hodgkin lymphomas, and extranodal NK/T cell lymphoma [380,381,382,383]. Glutaminase, specifically glutaminase 1 (GLS1) expressed in mitochondria, converts glutamine to glutamate [384], but also stimulates the growth of tumour cells [385] and is involved in autophagy [386], signal transduction [387], and radioresistance [388]. Recent evidence emerged showing that GLS1 might be involved in tumorigenesis and progression of human cancers [370]. GLS1 is overexpressed in metastatic lymph nodes and colorectal cancer cells [370]. Methionase, also known as L-Methionine-γ-lyase (EC 4.4.1.11; MGL), methioninase, L-methionine-γ-demethiolase, and L-methionine methanethiol-lyase (deaminating), is produced by *Pseudomonas putida*, *Pseudomonas ovalis*, *Micrococcus luteus*, *Arthrobacter* spp., *Corynebacterium glutamicum*, *Staphylococcus equorum*, *Citrobacter* spp., *Clostridium sporogenes*, *Trichomonas vaginalis* and *Entamoeba histolytica*, but has also been isolated from protozoans, fungal, archaeon, and plants [371]. Melignant tumours are highly dependent on methionine. Depletion of methionine through methionase-based therapy may be an important strategy to control the growth of cancer cells. One approach experimented with was linking L-methionase to human annexin-V to generate a fusion protein. The fusion protein catalyzed the conversion of nontoxic prodrug selenomethionine into toxic methylselenol and restricted the growth of tumour cells by depriving the cells from access to methionine [389,390,391]. The advantage of using the fusion protein is that it does not require to be delivered directly to the tumour cells but only to the bloodstream.

## 12. Conclusions and Future Directions

Cancer is a major health concern and treatment remains a challenge due to cells developing resistance. Despite numerous efforts to use viable bacteria in the treatment of cancer, the idea is still viewed with scepticism, as many strains experimented with are considered obsolete or opportunistic pathogens. To date, bacterial anticancer compounds studied were either toxins, antibiotics, bacteriocins, non-ribosomal peptides, polyketides, phenylpropanoids, phenylflavonoids, purine nucleosides, short chain fatty acids (SCFAs) or enzymes. Other natural antitumor compounds discovered are spliceostatins, such as spliceostatin B purified from cell-free extracts of *Pseudomonas* sp. 2663 and pladienolide B, a macrolide produced by *Streptomyces platensis* Mer-11107. More research on spliceostatins is required. It is interesting to note that many of the bacteria that produce anticancer compounds are either obligate or facultative anaerobes. Many anticancer compounds naturally produced by bacteria are specific in their mode of action, and some are selective in attacking only cancerous cells. Lactic acid bacteria form a major part of the gut microbiome, yet the role they play in cancer treatment is ill researched. The combined use of natural anticancer compounds with conventional anticancer therapy warrants further research. Progress in metagenomics, proteomics, heterologous gene expressions and nanotechnology, combined with the use of artificial intelligence software such as AlphaFold 2 (https://alphafold.ebi.ac.uk/) and Chemistry42 (https://arxiv.org/abs/2101.09050, accessed on 10 May 2022), may lead to the discovery and design of novel anticancer molecules.

## Figures and Tables

**Figure 1 microorganisms-10-01733-f001:**
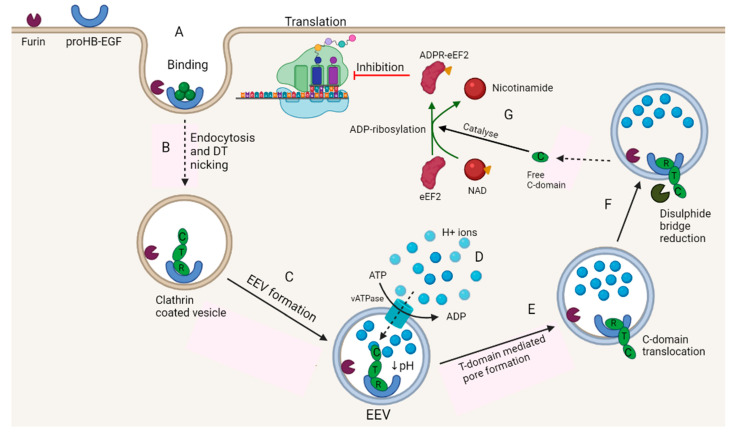
Mode of action of diphtheria toxin (DT). (A) The receptor (R domain) of DT, shown here as green speres, binds to the membrane-anchored precursor of heparin-binding epidermal-like growth factor (proHB-EGF). (B) DT is nicked and the DT-HB-EGF enters a clathrin-coated vesicle through endocytosis. Furin or furin-like proteases converts DT to mature form. (C) An early endosomal vesicle (EEV) is formed by replacing clathrin proteins with the GTPase Arf-1 and coat protein COPI (not shown). (D) EEV is acidified by the transport of protons (H^+^) across the membrane, instigated by vacuolar adenosine triphosphatase (vATPase). (E) The T-domain is translocated across the membrane, exposing the C-domain to the cytosol. (F) The disulphide bridge is reduced to liberate the catalytic C-domain. (G) The free C-domain catalyses the ADP-ribosylation of eukaryotic elongation factor 2 (eEF2) to ADPR-eEF2, which inhibits translation. This illustration was made using BioRender (https://biorender.com/, accessed on 12 May 2022).

**Figure 2 microorganisms-10-01733-f002:**
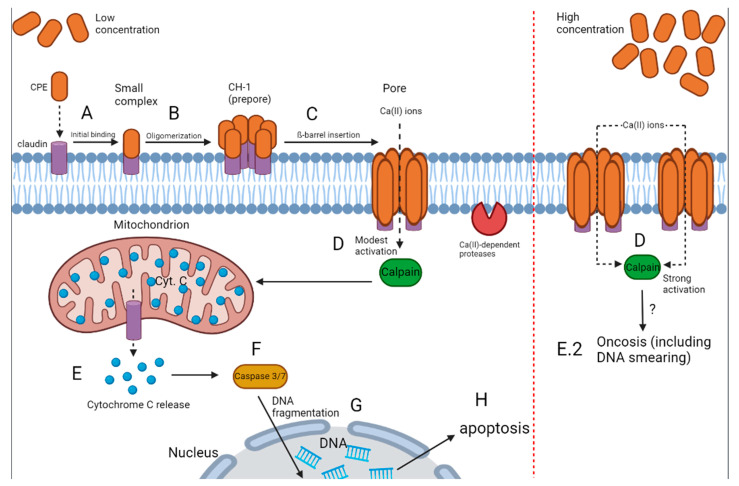
*Clostridium perfringens* enterotoxin (CPE) mode of action. (A) Tyrosine residues in the C-terminal of CPE interact with the second extracellular loop (ECL-2) of a receptor claudin (CLDN) and form small complexes. (B) Approximately six small complexes oligomerize to form a larger CPE hexamer 1 (CH-1*;* prepore). (C) β-hairpin loops of CPE assemble to form a β-barrel that inserts into the cell membrane to create a cation-permeating pore. (D) Influx of Ca^2+^ ions disrupts the osmotic equilibrium and activates Ca^2^-dependent proteases to lyse the cell that forms calpain. (E) Release of cytochrome C from the mitochondrion. (F) Activation of caspase 3/7 and formation of a large new CH-2 complex (approximately 600 kDa, consisting of CLDNs, occludins and the CPE hexamer. (G and H) Apoptosis, leading to DNA fragmentation. This illustration was constructed using BioRender (https://biorender.com/, accessed on 12 May 2022).

**Figure 3 microorganisms-10-01733-f003:**
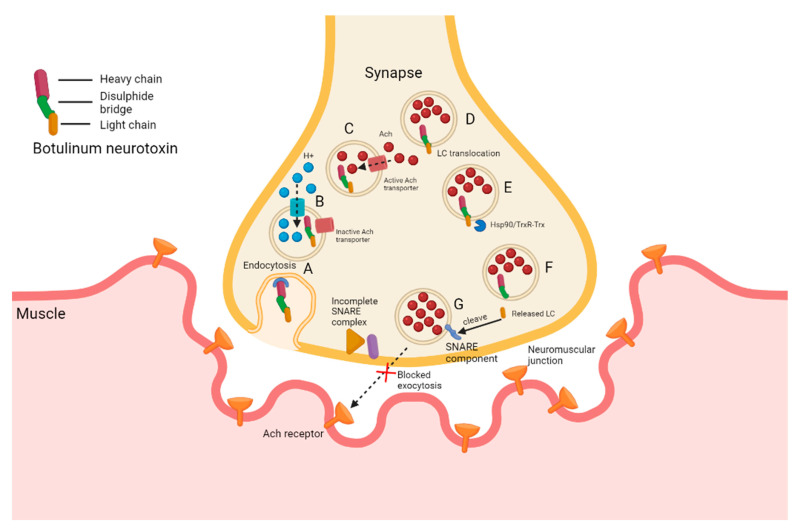
Botulinum neurotoxin (BoNT) mode of action. (A) BoNT binds to ecto-acceptors (polysialogangliosides) on the presynaptic cell surface of cholinergic neurons and is internalized via SV-2 or Syt-mediated endocytosis. (B) The synaptic vesicle is acidified with vesicular proton pumps, which in turn (C) activates Ach transporter proteins. The activated transporter proteins import acetylcholine (Ach) and the light chain (LC) of BoNT is translocated to the cytosol (D) with the heavy chain (HC) domain. (E) Heat shock protein 90 (Hsp90) and thioredoxin reductase-thioredoxin (TrxR-Trx) cleaves the LC and (F) liberates it into the cytosol. (G) The C-domain is a Zn^2+^-dependent endopeptidase that cleaves proteins in the soluble N-ethylmaleimide-sensitive adaptor receptor (SNARE) protein complex. This complex is responsible for exocytosis and the fusing of acetylcholine-containing vesicles with the plasma membrane, allowing the release of acetylcholine. The cleaved SNARE component is non-functional, thereby blocking the release of acetylcholine from the presynaptic membrane to muscles. Blocking exocytosis of acetylcholine leads to failed skeletal muscle contracture. This representation was constructed using BioRender (https://biorender.com/, accessed on 12 May 2022).

**Figure 4 microorganisms-10-01733-f004:**
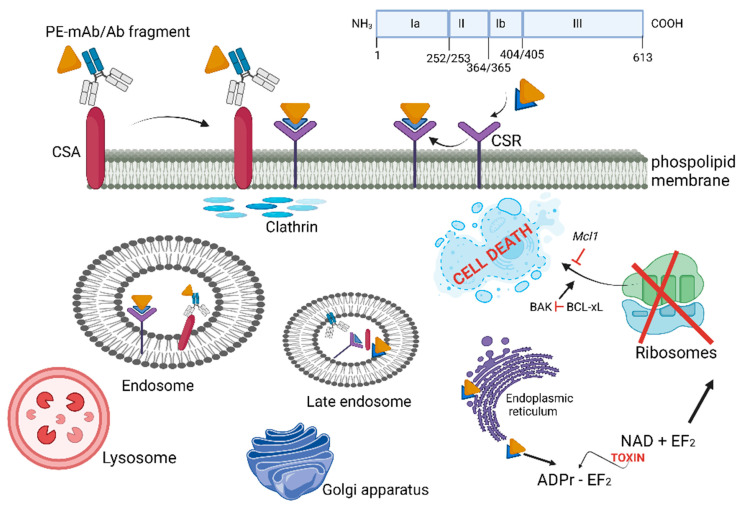
Top of presentation: The four domains (Ia, II, Ib and III) of *Pseudomonas* exotoxin (PE), with the sizes of each domain indicated in amino acid numbers. Interaction of PE-IT (immunotoxin) with a cancer-specific antigen (CSA) or cancer-specific receptor (CSR) on the surface of cancer cells. Intracellular events leading to cell death is illustrated below the phospholipid membrane. Abbreviations: PE-L = *Pseudomonas* exotoxin A, fused to a cancer-specific ligand; Ab = antibody; EF2 = eukaryotic elongation factor-2 on ribosomes; Mcl1 = gene encoding anti-apoptotic protein; BAK = Bak protein involved in mitochondrion outer membrane (MOM) permeabilization; BCL-xL = B-cell lymphoma-extra large that inhibits the activation of Bak, thereby preventing a loss of MOM integrity. This illustration was constructed using BioRender (https://biorender.com/, accessed on 15 August 2022).

**Figure 5 microorganisms-10-01733-f005:**
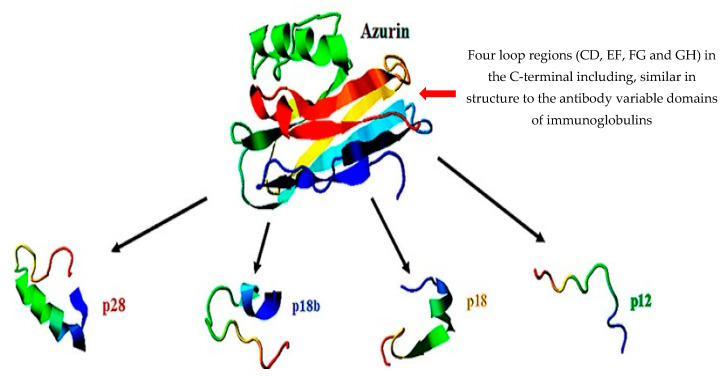
Azurin (128 amino acids), with an extended α-helix protein transduction domain, p28 (Leu50–Asp77), and four loop regions in its C-terminal (shown here in red, yellow, light blue and dark blue). The p28 peptide of 28 amino acids (amino acids 50 to 77) facilitates cell crossing and promotes apoptosis. Cell growth in tumours is repressed by 10 to 12 amino acids in the COOH terminal of p28. The α-helix, stretching over 18 amino acids (Leu^50^–Gly^67^), shown here as peptide p18, has a high affinity for cancer cells (less so for normal cells), excellent penetration abilities, and high binding to the tumour repressor protein p53. Peptide p18b also contains 18 amino acids (Val^60^–Asp^77^) but has a short α-helix and β-sheet and penetrates cancerous and normal cells. The p12 peptide of 12 amino acids (Gly^66^–Asp^77^) does not have an α-helical structure, binds poorly to p35, and penetrates cancer and normal cells. Adapted from Yaghoubi et al. [317].

**Figure 6 microorganisms-10-01733-f006:**
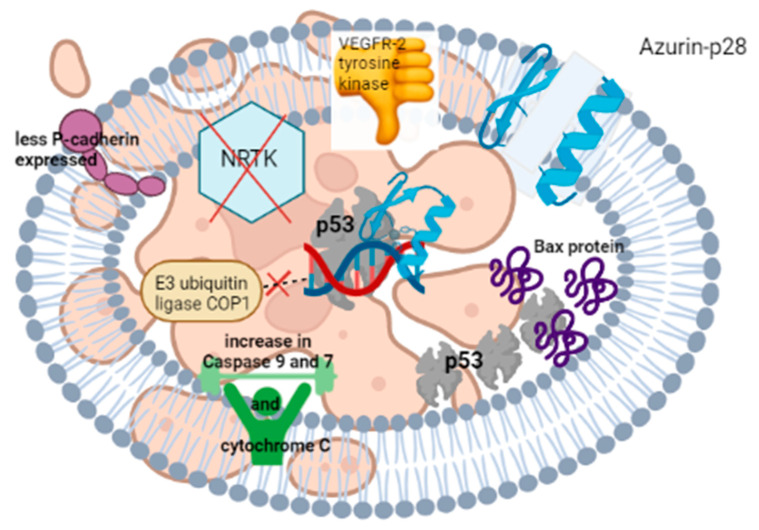
Mode of action of Azurin. Once inside a cancer cell, azurin attaches to the DNA-binding domain of the tumour-suppressor protein p53 (middle of presentation) and prevents binding of the latter to E3 ubiquitin ligase COP1, resulting in an increase in cytoplasmic p53 levels. Cell growth is repressed, and cells are destroyed by apoptosis. Azurin also interferes with non-receptor tyrosine kinase (NRTK) signalling pathways. Bax protein (a central cell death regulator) and cytochrome C levels increase, and caspases 9 and 7 are activated. VEGFR-2 tyrosine kinase activity is reduced, preventing the formation of new blood vessels and the expression of P-cadherin. Prepared using BioRender (https://biorender.com/, accessed on 12 May 2022).

**Table 1 microorganisms-10-01733-t001:** Humanized immunotoxins developed from fusion to diphtheria toxin (DT).

Immunotoxin	Toxin/Fragment	Targeting Moiety	Cancer or Cell Line	Result	Reference
Ontak	DT_389_	IL-2	Adult T-cell leukaemia and CTCL	Significant activity. FDA approval for CTCL treatment	[171]
mVEGF-DT	DT_386_	mVEGF	TC1-induced solid tumours	Tumour regression and an increase in survival rate	[174]
DTAT	C- and T-domains of DT	N-terminal of uPA	Glioblastoma cells	Selective killing and regression in tumour growth	[175]
DT_386_-BR2	DT_386_	Buforin II (BR2)	MCF-7 and HeLa cells K-562	Specific and significant reduction in survivability and apoptosis	[176,177]
Tagraxofusp (Elzonris^TM^)	DT_388_	IL-3	BPDCN, AML, CMML, & MM	FDA approval for BPDCN treatment	[178]
DT_389_GCSF	DT_389_	GCSF	HL-60	Specific apoptotic death and nuclease activity	[179]
hDT_806_	DT_390_	HuBiscFv806	4 HNSCC cell lines	Apoptosis, tumour size reduction and EGFR signalling disruption	[180]
PD1-DT	DT_386_	PD1	C57BL/6 tumorous mice	67% decrease in tumour volume	[181]
DT_389_-YP7	DT_389_	hYP7 scFv	HepG2 HCC	Decreased cell viability and specific toxicity	[182]

CTCL: cutaneous T-cell lymphoma; mVEGF: mouse vascular endothelial growth factor; DTAT: DT fused to the amino (N)-terminal of uPA; uPA: urokinase-type plasminogen activator; BPDCN: blastic plasmacytoid dendritic cell neoplasm; AML: acute myeloid leukaemia; CMML: chronic myelomonocytic leukaemia; MM: multiple myeloma; GCSF: granulocyte colony-stimulating factor; HuBiscFv806: humanized bivalent single-chain variable fragment of monoclonal antibody 806; HNSCC: head and neck squamous cell carcinoma; EGFR: epidermal growth factor receptor; PD1: programmed cell death protein-1; hYP7 scFv: humanized YP7 single-chain variable fragment; HCC: hepatocellular carcinoma.

**Table 2 microorganisms-10-01733-t002:** Studies using BoNT as an anti-cancer agent.

Cancer or Cell Line	Study Type	Methodology	Results	Reference
VCap cells Cancerous human prostate	In vivo In vivo	OnaA injection into VCap cells transplanted into murine prostate OnaA injection into prostate before prostatectomy	Inhibited cancer progression and increased apoptosis Increased incidence of apoptosis	[224]
MIA PaCa-2 cells	In vivo	Co-injection of cancer cells and 20 U/kg BoNT, or BoNT injection followed by cancer cell injection (murine study)	Increase in apoptosis and a decrease in tumour size	[225]
SiMA and SH-SY5Y cell lines	In vitro	BoNT/C injection into retinoic acid-treated	Increase in apoptosis	[223]
3T3 fibroblast cells	In vitro	BoNT/A treatment	Cytoplasmic degradation and decreased cell viability	[211]
SCC-25 and HUVEC cells	In vitro	Cells grown in the presence of BoNT	No effect on cell growth	[226]
DBTRG glioblastoma cell line	In vitro	BoNT/A and BoNT/A + AMG	Increased apoptosis and decreased cell proliferation	[227]

Abbreviations: OnaA: OnabotulinumtoxinA (Botox); DBTRG: Denver Brain Tumour Research Group; AMG: transient receptor potential vanilloid 1 receptor antagonist.

## Data Availability

Not applicable.

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
