# Peer review of "Do Bacteria Provide an Alternative to Cancer Treatment and What Role Does Lactic Acid Bacteria Play?"

_microorganisms, 2022, doi:10.3390/microorganisms10091733_

Round 1

Reviewer 1 Report

excellent work, thank you for the effort,  I enjoy reading it,  it helps me to understand the possible health benefits

Author Response

Dear reviewer

thank you for the kind comment and acceptance of the manuscript without further suggestions.

Yours sincerely

Prof LMT Dicks 

Reviewer 2 Report

Overall, I find that the paper is well-written, contains a number of interesting scientific facts, provides an extensive number of references, and is of general interest. However, I was not quite certain what scope was being reviewed and how it relates to the sub-theme of the role(s) of lactic acid bacteria in cancer. On one hand, live therapeutic bacteria such as William Coley's original work with Streptococcus, and later work on Clostridium and Salmonella are discussed, while other bacteria such as Listeria are not, nor are recent or current live bacterial cancer clinical trials included (e.g., Listeria immunotherapies by Advaxis; E. coli study by Synlogic; Clostridium study by BioMed Valley; Salmonella expressing IL-2 by Salspera). A revised version should include them. LAB make a number of different compounds with cancer therapeutic potential, have a probiotic preventative effect, and may have potential for therapeutic effect, but it's unclear if or why the authors are trying to discuss them separately rather than together with the other bacteria. In the Conclusions they are not even mentioned.  

Line 132 begins a discussion of Fusobacterium and Bacteriodies, which interfere with an antitumor immune response. However, there is no mention of the developments of how the microbiome (e.g., Bifidobacterium; mentioned above for its delivery capability) or specific subcomponents of organisms like Enterococcus prime the immune system, and even modulate the response to cancer chemotherapies.

Line 220-221. What was more cytotoxic to K562 cells?  

Line 226-227. Is aflatoxin produced in the gut and relevant to LAB suppressing cancer?  

In the bacterial toxins section, CPE is extensively reviewed, although it has never generated a useful drug, while Pseudomonas ToxA has led to the FDA-approved moxetumomab pasudotox - this should be included in the review.  

In the Bacteriocins section starting Line 644. The Gram-positive bacteriocins are defined by class, but there is no definition of the Gram-negative bacteriocins or colicins, which are usually defined by their mechanism of action.  

In the Enzymes section starting Line 945. The bacterial enzymes  glutaminase and methionase are not included.

Conclusions starting line 966. This is more forward looking so perhaps re-title Conclusions and Future Directions.

Lines 978 and 979 - it is a vast overstatement to say bacterial products only kill cancer cells, although some are selective.

Minor corrections:

Line 2. Title: Change Does to Do.

Line 105: Change proofed to proved.

Line 173: Change comma to period after [85] or rewrite sentence to clarify.  

Author Response

Dear Reviewer

Thank you for the valuable comments. All have been addressed and changes to the text have been highlighted.

Comment:

Overall, I find that the paper is well-written, contains a number of interesting scientific facts, provides an extensive number of references, and is of general interest. However, I was not quite certain what scope was being reviewed and how it relates to the sub-theme of the role(s) of lactic acid bacteria in cancer. On one hand, live therapeutic bacteria such as William Coley's original work with Streptococcus, and later work on Clostridium and Salmonella are discussed, while other bacteria such as Listeria are not, nor are recent or current live bacterial cancer clinical trials included (e.g., Listeria immunotherapies by Advaxis; E. coli study by Synlogic; Clostridium study by BioMed Valley; Salmonella expressing IL-2 by Salspera). A revised version should include them. LAB make a number of different compounds with cancer therapeutic potential, have a probiotic preventative effect, and may have potential for therapeutic effect, but it's unclear if or why the authors are trying to discuss them separately rather than together with the other bacteria. In the Conclusions they are not even mentioned.  

Answer:

The review summarises the importance of bacteria and their metabolites in the treatment of cancer.  Recently, several papers have been published on the role lactic acid bacteria (LAB), and their metabolites, play in cancer, and the treatment thereof.  Since the anticarcinogenic properties of lactic acid bacteria (LAB) is addressed in far fewer publications and focuses mainly on exopolysaccharides (EPS), peptidoglycan, nucleic acid, bacteriocins, and S-layer proteins (as stated in lines 214-216), a review on the topic is, we believe, is pertinent and timely.

We have included additional information on Salmonella (lines 119-126), Clostridium (lines 132-154), Listeria (lines 155-175) and E. coli (lines 176-188) as requested but refrained from mentioning company names as this may be viewed as being biased.  We did, however, discuss the findings and importance of these studies and referenced the key papers, in which the names of companies are mentioned.  The paper has thus been revised to include all relevant scientific data.

We have, as far as possible, included the discussion on LAB in the same sections with other bacteria, under the same subheadings, e.g. section 2 (“Bacterial-mediated Cancer Therapy”) lines 214-300, 301-326, and 327-335; section 5 (“Bacteriocins”) lines 805-814; 897-905; 921-969; and section 10 (“Short Chain Fatty Acids”).

The conclusions has been amended by referring to lactic acid bacteria (lines 1127 and 1128).

Comment:

Line 132 begins a discussion of Fusobacterium and Bacteriodies, which interfere with an antitumor immune response. However, there is no mention of the developments of how the microbiome (e.g., Bifidobacterium; mentioned above for its delivery capability) or specific subcomponents of organisms like Enterococcus prime the immune system, and even modulate the response to cancer chemotherapies.

Answer:

The section has now been amended, as suggested, by adding a sentence (lines 189 and 190) and a section from lines 301 to 326.

Comment:

Line 220-221. What was more cytotoxic to K562 cells?  

Answer:

The sentence has been amended by referring to the six lactic acid bacteria (see line 278).

Comment:

Line 226-227. Is aflatoxin produced in the gut and relevant to LAB suppressing cancer?  

Answer: The sentence has been corrected by referring to aflatoxin production in food (line 284).

Comment:

In the bacterial toxins section, CPE is extensively reviewed, although it has never generated a useful drug, while Pseudomonas ToxA has led to the FDA-approved moxetumomab pasudotox - this should be included in the review.  

Answer:

A section on Pseudomonas aeruginosa toxin has now been included (lines 621-666, plus Figure 4).

Comment:

In the Bacteriocins section starting Line 644. The Gram-positive bacteriocins are defined by class, but there is no definition of the Gram-negative bacteriocins or colicins, which are usually defined by their mechanism of action.  

Answer:

A section addressing the classification of Gram-negative bacteriocins has been added (lines 788-794).

Comment:

In the Enzymes section starting Line 945. The bacterial enzymes  glutaminase and methionase are not included.

Answer:

The two enzymes have now been mentioned (line 1079) and additional information added (lines 1094-1112).

Comment:

Conclusions starting line 966. This is more forward looking so perhaps re-title Conclusions and Future Directions.

Answer:

The heading was changed to “Conclusions and Future Directions”.

Comment:

Lines 978 and 979 - it is a vast overstatement to say bacterial products only kill cancer cells, although some are selective.

Answer:

The sentence has been amended (lines 1125-1127).

Minor corrections:

Line 2. Title: Change Does to Do.

Line 105: Change proofed to proved.

Line 173: Change comma to period after [85] or rewrite sentence to clarify.  

Answer:

All these corrections have been made (lines 2, 105 and 190).

Yours sincerely

Prof LMT Dicks

Reviewer 3 Report

The abbreviation of bacteria and some characters are not consistent.

The role of lactic acid bacteria is not significant.

The concept of the figure needs to be interjected throughout the review. The figure is only mentioned lately on in the review, and then it is essentially forgotten.

The review could benefit from another figure to help further develop concepts.

Author Response

Dear Reviewer

Thank you for the comments. These have been addressed and highlighted in the text.

Comment:

The abbreviation of bacteria and some characters are not consistent.

Answer:

All abbreviations have been checked.

Comment:

The role of lactic acid bacteria is not significant.

Answer:

The review summarizes the importance of bacteria and their metabolites in the treatment of cancer.  Recently, several papers have been published on the role lactic acid bacteria (LAB), and their metabolites, play in cancer, and the treatment thereof.  Since the anticarcinogenic properties of lactic acid bacteria (LAB) is addressed in far fewer publications and focuses mainly on exopolysaccharides (EPS), peptidoglycan, nucleic acid, bacteriocins, and S-layer proteins (as stated in lines 214-216), a review on the topic is, we believe, is pertinent and timely.

Comment:

The concept of the figure needs to be interjected throughout the review. The figure is only mentioned lately on in the review, and then it is essentially forgotten.

Answer:

The figures have been discussed and referenced in the appropriate sections.

Comment:

The review could benefit from another figure to help further develop concepts.

Answer:

Figure 4 has been added, bringing the total to six Figures.

Yours sincerely

Prof LMT Dicks

Round 2

Reviewer 3 Report

We are satisfied with the revised manuscript.